# ONCE-MORE: CONTINUOUS SELF-CORRECTION FOR LARGE LANGUAGE MODELS VIA PERPLEXITY-GUIDED INTERVENTION

**Jiaxun Gao**[*]     **Him Wai Ng**[*]     **Z. Jane Wang**

Department of Electrical and Computer Engineering
University of British Columbia
`jiaxun.gao@ece.ubc.ca`   `michael.ng@ubc.ca`   `zjanew@ece.ubc.ca`

## ABSTRACT

Large Language Models (LLMs) often experience compounding errors during long text generation. Early errors could propagate and lead to drift, faulty reasoning, or repetition. Self-correction is a promising technique for addressing this issue. However, the existing main approaches have limitations. Supervised training methods can build self-correcting behaviours into models, but require training data collection and lack cross-domain generalizability. Current post-hoc iterative refinement methods operate only at inference time, but have to wait for substantial portions of the draft to be generated before providing feedback. Such feedback can not guarantee effective guidance, and the same error patterns can reappear. In this paper, we propose Once-More, a model-agnostic post-hoc self-correction framework that intervenes during generation. Once-More leverages token-level perplexity and verifier feedback to provide continuous, guided steering of the generation path through a logit suppression mechanism, and therefore helps accumulate "more correct" steps throughout the generation process. Evaluation on multiple benchmarks demonstrates that the proposed Once-More achieves state-of-the-art results when compared to representative self-correction methods. To our knowledge, Once-More is the first post-hoc method to leverage token perplexity and external feedback for continuous, guided self-correction.

## 1 INTRODUCTION

Large Language Models (LLMs) have demonstrated remarkable capabilities in text generation and complex reasoning tasks (Sun et al., 2025; Ferrag et al., 2025). However, their autoregressive nature poses a fundamental challenge: during generation, LLMs can produce errors or inaccuracies that further propagate through subsequent tokens, making errors to compound and eventually drive the generation away from the target (Arbuzov et al., 2025; Wang et al., 2023). This error-compounding issue raises serious concern when deploying LLMs in critical decision-making processes. Many LLM-based agentic workflows suffer from reliability issues stemming from this concern, which limits their viability for real-world deployment. (Pan et al., 2025; Gabison & Xian, 2025).

To address such concerns, researchers have developed various self-correction methods which enable LLMs to modify or revise their outputs at inference (Kamoi et al., 2024; Pan et al., 2024). Such methods identify errors during generation and guide models toward better responses through targeted feedback mechanisms (Paul et al., 2023; Tyen et al., 2023; Shinn et al., 2023). The feedback on the final output is provided via external knowledge bases (Gou et al., 2024; Zheng et al., 2024) or by more capable models (e.g., GPT-4) (Koutcheme et al., 2024). However final-output feedback is too coarse to provide effective steering, and thus these methods often fail to prevent recurring error patterns and lead to non-convergent refinement loops (Kamoi et al., 2024; Xu et al., 2024). Some methods use supervised fine-tuning as an alternative to build self-correction behaviour directly into models (Yan et al., 2025; Liu et al., 2025; Wang et al., 2024a); however, they are inherently

---

[*]Co-first authors. Listed alphabetically by surname.

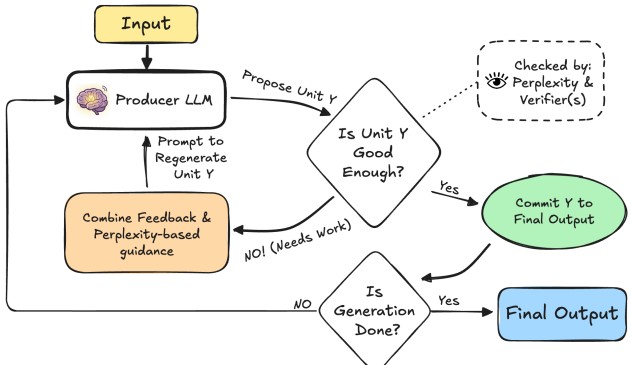

Figure 1: Overview of the proposed Once-More framework (Sec. 3.1). The Producer generates adaptive units, which are checked by Perplexity and Verifier(s). Rejected units trigger regeneration via the combined feedback and perplexity-driven guidance, creating a continuous self-correction process that intervenes before errors propagate.

limited by the distribution of their training data. They are susceptible to performance degradation on out-of-distribution tasks, where uncorrected errors can cascade through the generation.

Rather than building domain-specific models with built-in self-correction, we believe that iterative refinement remains the most general and practical approach. However, it requires fundamental improvements in guidance granularity. Effective self-correction should function continuously throughout generation, progressively steering models along better trajectories, and eventually allow "more correct" incremental decisions to accumulate into better final outputs. Motivated by this vision, in this paper, we propose an inference-time framework, **Once-More**, which performs continuous self-correction on any LLM generation process. It transforms the LLM generation process into a multi-agent interaction process: a Producer generates content while Verifier(s) evaluate and provide continuous guidance. Once-More operates at the level of units of generation (e.g., clauses, sentences, paragraphs, or code blocks). A complete generation is a sequential composition of unit generations. When the Producer generates a unit, the framework computes its perplexity to assess potential errors. High perplexity triggers verification for goal alignment, correctness, and consistency. Rejected units are regenerated using evaluation feedback combined with perplexity-driven logit redistribution until the Verifier(s) accept the unit. Figure 1 shows the high-level overview of the proposed Once-More. The framework also supports LLM, non-LLM, or tool-augmented Verifiers.

The proposed Once-More are evaluated on various benchmarks, including Olympiad mathematics (AIME 2024 (Art of Problem Solving, 2024a;b) and 2025 (Art of Problem Solving, 2025a;b)), graduate-level science questions (GPQA) (Rein et al., 2024), LiveBench (reasoning) (White et al., 2024), SVAMP (Patel et al., 2021), and GSM8K (Cobbe et al., 2021). It achieves state-of-the-art performance when compared to other self-correction methods. Our contributions are as follows: (1) We propose the Once-More framework to perform continuous, fine-grained self-correction, which yields State-of-the-Art performance while being more token-efficient than previous iterative refinement methods; (2) We present a method to uniquely fuse perplexity-based uncertainty signals with external verifier feedback. It is employed in Once-More to prevent repeated errors while preserving model beliefs during generation. To our knowledge, the proposed Once-More is the first post-hoc method to leverage token perplexity and external feedback for continuous, guided self-correction; (3) Our proposed continuous guidance mechanism in Once-More enables LLMs to perform reasoning-like corrections dynamically during generation, which helps to detect early errors and move to better generation trajectories.

## 2 RELATED WORK

**Error accumulation in LLMs**. The problem of error propagation in autoregressive models has been extensively studied. Classical analyses of exposure bias demonstrate that models trained on teacher-forced prefixes often struggle at test time, leading to error compounding in long generations

(Arora et al., 2022; Schmidt, 2019). LLMs also suffer from the error accumulation issue, although LLM's error accumulation patterns are more nuanced than simple exponential decay (Arbuzov et al., 2025). While scaling up LLMs enables the emergence of self-correction behaviours (Liu et al., 2024; Wang et al., 2024b), they remain unpredictable and uncontrollable at inference. To address this concern, we propose making self-correction explicit and controllable through perplexity-guided intervention. Our proposed approach is model-agnostic and can be used with smaller-sized LLMs, making it more practical for local or edge use cases.

**Self-correction via iterative refinement**. Several approaches can enable LLMs to revise their outputs through iterative processes. For instance, CRITIC uses external tools to verify model outputs and provides corrective feedback (Gou et al., 2024), Self-Refine employs models as both generators and evaluators (Madaan et al., 2023), and Verify-and-Edit incorporates knowledge bases for factual correction (Zhao et al., 2023). While these approaches can improve response quality, they operate on completed drafts or coarse-grained steps, and therefore could allow errors to compound before intervention occurs. The prompt-only feedback also occasionally fails to prevent recurring error patterns. Our proposed Once-More addresses the above concerns by intervening at a more granular level with continuous monitoring, enabling more direct and localized feedback that has a more effective impact on the model's generation process.

**Self-correction via supervised fine-tuning**. Another line of work builds self-correction behaviours directly into the models through training. For instance, $S^3$c-MATH trains spontaneous self-correction for mathematical reasoning (Yan et al., 2025), LLaVA-SCo extends this to vision-language models (Liu et al., 2025), and the learning-from-failure approach finetunes models to internalize correction behaviours (Wang et al., 2024a). However, their gains remain modest, and these methods require extensive training data and are susceptible to performance degradation on out-of-distribution tasks. To tackle such issues, our proposed framework remains model-agnostic and training-free, while providing continuous guidance for self-correction behaviours.

**Multi-agent and role-playing approaches**. Formulating generation as a multi-agent interaction has been shown to be effective for complex tasks. E.g., Reflexion equips agents with verbal feedback and episodic memory to improve over trials (Shinn et al., 2023), MAgICoRe combines solver, reviewer, and refiner roles with reward models for targeted step-wise error correction (Chen et al., 2024), and ReAct interleaves reasoning with external actions to reduce hallucinations (Yao et al., 2023). These methods leverage role specialization to provide feedback in different perspectives, and inspired by their success, our proposed Once-More adopts this idea in its Producer-Verifier architecture. However, their coarse-grained feedback and prompt-only interactions cannot guarantee effective steering. We therefore plan to enhance the multi-agent system with continuous monitoring and direct probability intervention.

**Decoding time steering**. The approach that manipulates sampling at inference time offers another avenue for controlling generation behaviour. E.g., GeDi guides generation toward desired attributes with discriminative models (Krause et al., 2020), Contrastive Decoding reduces hallucination by subtracting weaker model logits (Li et al., 2022; O'Brien & Lewis, 2023), and DoLa sharpens factuality through layer-wise contrast (Chuang et al., 2023). Recent work on Cautious Next Token Prediction demonstrates that perplexity-based mechanisms can effectively guide generation decisions (Wang et al., 2025b), providing a theoretical basis for our approach. However, existing methods only operate globally or target high-level attributes, lacking fine-grained control. Inspired by these methods, our proposed method manipulates sampling at inference time, but also employs localized logit manipulation to perform a more effective generation steering.

# 3 ONCE-MORE FRAMEWORK

We now present Once-More, a model-agnostic framework that performs continuous self-correction during LLM generation. The key idea is simple: rather than waiting for a complete output before providing feedback, we monitor generation quality at intermediate steps and intervene immediately when potential errors are detected. This section describes the framework's architecture (Sec. 3.1), how it monitors generation quality (Sec. 3.2), and how it guides corrections (Sec. 3.3).

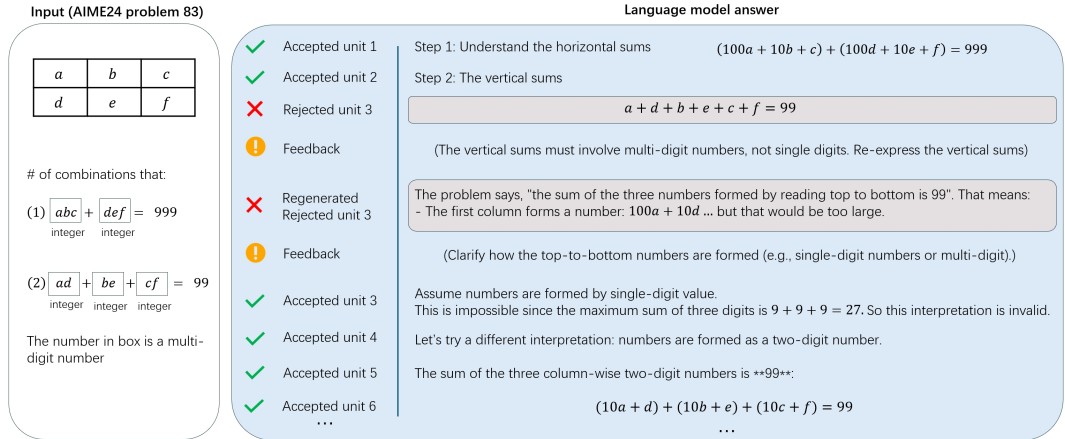

Figure 2: Illustration example: The proposed Once-More corrects a mathematical interpretation error on AIME 2024 Problem 83. It rejects an incorrect single-digit interpretation (unit 3), provides feedback, and guides the model to reconsider. The model then explicitly reasons why the initial is impossible and pivots to the correct two-digit interpretation, demonstrating intermediate reasoning.

## 3.1 FRAMEWORK ARCHITECTURE

As shown in Figure 1, Once-More transforms standard LLM generation into a monitored multi-agent process with three core components:

**Producer.** A frozen, pre-trained LLM that generates content incrementally in *units* (adaptive chunks such as sentences, paragraphs, or code blocks). The Producer operates at inference time without any training or fine-tuning. The only requirement is access to token-level probability scores during decoding, which many providers already expose through their APIs. This requirement matters only for our regeneration mechanism and doesn't constrain the choice of model architecture.

**Verifier(s).** One or more agents that evaluate each provisional *unit* against the task **Goal** and constraints, returning a binary decision (accept or reject) along with optional feedback $F$ in natural language or structured format. Verifiers may be LLMs, non-LLM programs, or tool-augmented modules as in tool-using agents (Yao et al., 2023; Shen et al., 2023; Shen, 2024).

**Generation Units.** Rather than working with fixed token windows, Once-More adapts its intervention granularity to the task context. For mathematical reasoning, a unit might be a single equation or derivation step; for code generation, a function or block; for prose, a sentence or paragraph. Units are identified using syntactic markers (punctuation, indentation) or learned boundary predictors. This adaptivity enables intervention at the most appropriate granularity for each task.

**The Generation-Verification Loop.** The proposed Once-More framework operates through continuous generation, monitoring, and correction loops. Figure 2 demonstrates this on a mathematical problem, showing how the framework catches and corrects an incorrect interpretation mid-solution. Let **Goal** denote the task and **Context** the generation history that has been verified and accepted. The framework operates through the following loop:

1. **Generate:** The Producer generates a provisional unit $Y = [y_1, \ldots, y_n]$ conditioned on (Goal, Context), where $y_i$ is one output token at time $i$.

2. **Monitor:** Compute the unit's perplexity $\mathrm{PPL}_{\mathrm{unit}}(Y)$ as an uncertainty signal (Sec. 3.2).

3. **Decide:**
   - If $\mathrm{PPL}_{\mathrm{unit}}(Y) \leq P_{\mathrm{th}}$ (low uncertainty), trust the unit and append to Context.
   - If $\mathrm{PPL}_{\mathrm{unit}}(Y) > P_{\mathrm{th}}$ (high uncertainty), invoke Verifier for explicit checking.

4. **Verify & Correct:**

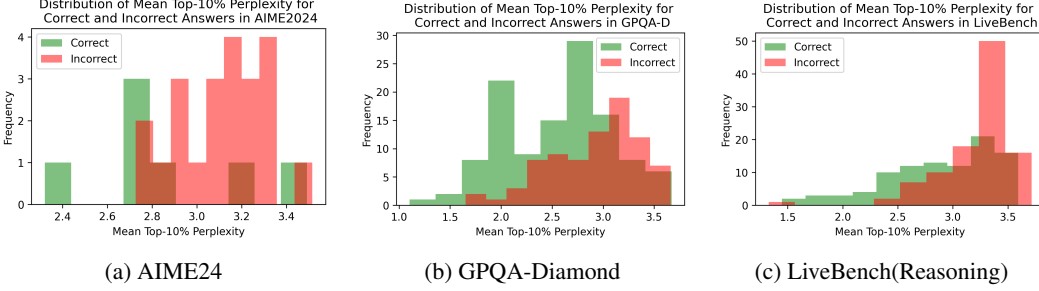

|   |   |   |
|:---:|:---:|:---:|
| (a) AIME24 | (b) GPQA-Diamond | (c) LiveBench(Reasoning) |

Figure 3: Distributions of the mean top-10% perplexity for correct vs. incorrect answers across benchmarks. Incorrect units consistently exhibit higher perplexity.

- If accepted: Append to Context and create a checkpoint.
- If rejected: Trigger guided regeneration (Sec. 3.3) using verifier feedback and probability adjustments.
- If regeneration also fails: Roll back to the previous checkpoint and regenerate from there, as the error may have originated earlier.

## 3.2 MONITORING GENERATION QUALITY VIA PERPLEXITY

To detect potential errors during generation, we need a real-time quality signal that doesn't require external verification for every token. We leverage *perplexity*, which is a standard language modelling metric that measures prediction uncertainty (Jelinek et al., 1977; Christopher Manning, 2021; Hugging Face, 2025). Recent work shows that entropy-based metrics provide reliable signals for response quality (Fu et al., 2025; Yang et al., 2025b; Wang et al., 2025a), which motivates our approach.

**Intuition.** When a model confidently predicts the next token (high probability on one candidate), perplexity is low. When probability mass spreads across multiple candidates, perplexity increases, signalling uncertainty. High uncertainty often correlates with potential errors, indicating the model is likely unsure what to generate next, and it's probably on the wrong path.

**Token-Level Perplexity.** At each position $t$, the Producer outputs a probability distribution $q_t(v)$ over the vocabulary $\mathcal{V}$. Rather than computing perplexity over the entire vocabulary (expensive), we approximate using the top-$K$ most likely tokens $S_t = \{v_{t,1}, \ldots, v_{t,K}\}$ (Holtzman et al., 2018; Fan et al., 2018):

$$\text{PPL}_t^{(K)} \;=\; \exp\!\left(\frac{1}{K}\sum_{i=1}^{K}(-\log q_t(v_{t,i}))\right) \tag{1}$$

This efficiently captures local uncertainty: $\text{PPL}_t^{(K)} \approx 1$ when one token dominates, and increases as mass spreads across alternatives.

**Unit-Level Aggregation & Verification Trigger.** For a unit $Y = [y_1, \ldots, y_n]$, we average token-level perplexities:

$$\text{PPL}_{\text{unit}}(Y) \;=\; \frac{1}{n}\sum_{t=1}^{n}\text{PPL}_t^{(K)} \tag{2}$$

Verification is triggered when $\text{PPL}_{\text{unit}}(Y) > P_{\text{th}}$, where $P_{\text{th}}$ is calibrated on a small held-out set to target a desired verification rate (e.g., check the top 25% most uncertain units). Details of the calibration procedure are in Appendix A.2. Figure 3 empirically validates this approach: on our benchmarks, incorrect units consistently show substantially higher perplexity than correct ones, demonstrating clear separation between reliable and problematic generations.

### 3.3 Guided Regeneration via Probability Adjustment

When a unit is rejected, simply asking the model to "try again" often produces the same mistake, since the model's learned biases can lead it down the same path (Xu et al., 2024). The model may acknowledge the feedback but still sample the same tokens due to strong learned priors. To break this cycle, Once-More performs *guided regeneration*: it (1) incorporates verifier feedback into the prompt, and (2) directly adjusts the token probability distribution to suppress previously chosen tokens and explore alternatives.

**Overview of the Adjustment Mechanism.** The adjustment process operates at the token level during regeneration:

1. **Identify high-uncertainty tokens:** Convert the rejected unit's token-level perplexities into position-wise suppression strengths; therefore, tokens with higher uncertainty receive stronger suppression.
2. **Align attempts:** Match tokens between the rejected attempt and the current regeneration attempt to determine which tokens to suppress at each position.
3. **Redistribute probability mass:** At each position, decrease the probability of the previously chosen token and proportionally increase alternatives, ensuring a valid probability distribution.

We now detail each step.

**Step 1: Converting Perplexity to Suppression Strength.** Let $[x_1, \ldots, x_{n_-}]$ denote the rejected unit with token-level perplexities $\{\text{PPL}_i\}_{i=1}^{n_-}$. We normalize these into suppression weights:

$$\hat{u}_i^{(1)} = \frac{\text{PPL}_i - \min(\text{PPL})}{\max(\text{PPL}) - \min(\text{PPL}) + \varepsilon} \in [0, 1] \tag{3}$$

where $\varepsilon$ is a small constant for numerical stability. This maps perplexity to a [0,1] scale: tokens with high perplexity (high uncertainty) get suppression weights near 1, while confident tokens get weights near 0.

**Step 2: Aligning Regeneration with Previous Attempt.** When regenerating, the new token sequence $[y_1, \ldots, y_n]$ may differ in length and content from $[x_1, \ldots, x_{n_-}]$. We establish a monotone alignment to transfer suppression weights: for each position $j$ in the new attempt, find the first matching token $x_i$ from the previous attempt (if any), creating an alignment matrix $A \in \{0, 1\}^{n_- \times n}$. This identifies which token should be suppressed at each regeneration position:

$$\text{target}_j = \begin{cases} x_i, & \text{if a match exists at position } i \\ x_j, & \text{otherwise (use position-based fallback)} \end{cases} \tag{4}$$

To avoid over-suppressing tokens that matched only by coincidence (e.g., common words appearing at distant positions), we apply distance decay:

$$\hat{u}_j^{\rightarrow} = A_{ij} \cdot \exp\left(-\left(\frac{|i-j|}{\tau}\right)^{\gamma}\right) \cdot \hat{u}_i^{(1)} \tag{5}$$

where $\tau$ controls the decay rate and $\gamma \in \{1, 2\}$ the decay shape. This reduces suppression strength for matches far from their original position.

We further apply Gaussian smoothing to prevent over-localization, spreading suppression signals across nearby positions (details in Appendix A.3). The final effective suppression at position $j$ is:

$$\alpha_j = \alpha \cdot u_j^{\star}, \quad \alpha \in (0, 1) \tag{6}$$

where $u_j^{\star}$ blends the direct and smoothed suppression signals, and $\alpha$ is a global scaling factor.

**Step 3: Probability Redistribution.** Let $q_j^{(2)}(v)$ denote the Producer's original next-token distribution at position $j$ during regeneration. We adjust this distribution to suppress the target token while boosting alternatives:

$$s_j(v) = \begin{cases} 1 - \alpha_j \, r_j(v), & v = \text{target}_j \quad \text{(suppress)} \\ 1 + \kappa_j \, r_j(v)^{\beta}, & v \neq \text{target}_j \quad \text{(boost)} \end{cases} \tag{7}$$

where $r_j(v) = q_j^{(2)}(v) / \max(\varepsilon, \bar{q}_j^{(1)}(v))$ measures how much the model's belief about token $v$ has shifted between attempts. The coefficient $\kappa_j$ is computed to preserve probability mass:

$$\kappa_j \;=\; \frac{\alpha_j \, r_j(\text{target}_j) \, q_j^{(2)}(\text{target}_j)}{\displaystyle\sum_{u \neq \text{target}_j} q_j^{(2)}(u) \, r_j(u)^{\beta}} \tag{8}$$

ensuring that the total probability removed from the target equals the total added to alternatives (derivation in Appendix A.4). The parameter $\beta \geq 0$ controls how redistributed mass is allocated and is set to 1 for all our experiments.

The final adjusted distribution is:

$$\tilde{q}_j(v) \;=\; \frac{q_j^{(2)}(v) \, s_j(v)}{\sum_{u \in \mathcal{V}} q_j^{(2)}(u) \, s_j(u)} \tag{9}$$

This is a valid probability distribution (non-negative, sums to 1) that explores alternatives while respecting the model's learned beliefs.

**Regeneration Process.** With the adjusted distribution $\tilde{q}_j(\cdot)$, the Producer samples a new unit. The verifier feedback is also appended to the prompt context, providing semantic guidance alongside the probability adjustment. If this regenerated unit is accepted, it becomes the new checkpoint. If rejected again, the framework rolls back to the previous checkpoint (before the problematic unit) and regenerates from there, recognizing that the error may have originated earlier in the generation. By combining perplexity-driven uncertainty detection with probability-level intervention, Once-More achieves continuous, fine-grained self-correction that prevents error accumulation during generation.

## 4 EXPERIMENTS

### 4.1 SETUP AND BASELINES

Due to the rapid development of the LLM field, prior self-correction studies employ diverse experimental setups across different models and benchmarks. To establish reproducible comparisons, we selected the popular open-sourced Qwen3 family of models (4B, 8B, and 14B parameters) as our base models (non-thinking) (Yang et al., 2025a). These model sizes were chosen for three key reasons: (1) they span small to medium scales, revealing how self-correction scales with capacity; (2) they represent practical compute-constrained scenarios; (3) Qwen3 offers state-of-the-art performance at these sizes, ensuring our results are relevant to current practice.

We evaluated Once-More on six benchmarks: AIME 2024 (Art of Problem Solving, 2024a;b) and 2025 (Art of Problem Solving, 2025a;b) for complex mathematical reasoning, LiveBench (reasoning subset) (Rein et al., 2024) for general reasoning, GPQA Diamond (Rein et al., 2024) for graduate-level science questions, and SVAMP (Patel et al., 2021) and GSM8K (Cobbe et al., 2021) for arithmetic word problems. This benchmark selection covers diverse reasoning types and difficulty levels.

We compare the proposed Once-More with four representative baselines:

- **Raw**: Direct generation without self-correction.
- **Self-Refine** (Madaan et al., 2023): The most widely-cited iterative refinement method, representing prompt-based self-correction approaches.
- **CRITIC** (Gou et al., 2024): Incorporates external tools for verification, representing tool-augmented self-correction methods and most recent post-hoc method.
- **S³c-MATH** (Yan et al., 2025): A supervised fine-tuning approach, representing training-based self-correction methods.

Here, we implemented the Self-Refine and CRITIC baselines according to their references. For Self-Refine, the number of refinement iterations was set to $k = 3$. For CRITIC, since no suitable resources were specified in the original work, we adapted the method by leaving the external evidence

Table 1: Accuracy performance (mean accuracy % ± std. dev. over 3 runs) comparisons between the proposed Once-More and baseline self-correction methods across multiple benchmarks. Results are for Qwen3 models of varying sizes (4B, 8B, 14B parameters). Once-More consistently outperforms iterative refinement approaches (Self-Refine, CRITIC) and raw generation.

|  |  | AIME24 | AIME25 | LiveBench(Reason.) | GPQA Diamond |
|---|---|---|---|---|---|
| Qwen3 4B | Raw | 13.3 ±3.3 | 20.0 ±3.3 | 20.3 ±1.2 | 43.9 ±1.3 |
|  | SelfRefine | 13.3 ±3.3 | 18.9 ±3.8 | 21.0 ±0.9 | 43.9 ±1.0 |
|  | CRITIC | 14.4±1.9 | 20.0±3.3 | 20.7±1.3 | **48.9**±0.8 |
|  | ours | **16.7**±3.4 | **23.3**±3.4 | **33.0**±1.0 | 47.5±0.9 |
| Qwen3 8B | Raw | 24.4±1.9 | 16.7±3.4 | 30.5±0.9 | 45.4±0.9 |
|  | SelfRefine | 25.6±3.9 | 15.5±3.9 | 31.2±1.3 | 47.0±1.7 |
|  | CRITIC | 24.4±1.9 | 18.8±3.9 | 31.5±1.3 | 48.9±1.5 |
|  | ours | **33.3**±3.3 | **24.4**±1.9 | **39.5**±1.3 | **49.5**±1.7 |
| Qwen3 14B | Raw | 26.7±3.3 | 18.8±3.8 | 44.0±1.3 | 48.0±1.5 |
|  | SelfRefine | 28.8±3.8 | 23.3±3.3 | 46.5±1.5 | 49.2±1.0 |
|  | CRITIC | 28.8±1.9 | 22.2±3.9 | 45.5±1.5 | 50.5±1.3 |
|  | ours | **36.7**±3.3 | **26.7**±3.3 | **52.5**±1.8 | **55.6**±1.6 |

Table 2: Accuracy performance comparisons with supervised fine-tuning methods on mathematical benchmarks. Once-More achieves competitive or superior performance without any training.

| Benchmark | Llama 3 8B | | | | Qwen2 Math 7B | | | |
|---|---|---|---|---|---|---|---|---|
|  | MetaMath SFT | S$^3$C w/o R&I | S$^3$C MathQA | Ours | MetaMath SFT | S$^3$C w/o R&I | S$^3$C MathQA | Ours |
| SVAMP | 81.4 | 80.5 | 81.8 | **82.0** | 85.6 | 86.7 | 87.4 | **89.0** |
| GSM8K | 81.1 | 81.7 | **82.9** | 79.0 | 84.1 | 84.4 | 84.8 | **85.2** |

field empty in math experiments. On the GPQA Diamond benchmark, CRITIC uses the first 1000 words from web search results as external evidence. S$^3$c-MATH results are taken from their published papers on comparable models. All reported results represent means over three independent runs. For Once-More's default configuration, we set $K = 5$ for top-$K$ perplexity calculation, with suppression factor $\alpha = 1.0$, redistribution sharpness $\beta = 1.0$, distance decay $\tau = 1$, diffusion bandwidth $\sigma = 1$ and perplexity threshold $\eta = 25\%$ fixed. The verifier feedback is generated from an agent using the same LLM as the producer agent across all experiments.

## 4.2 MAIN RESULTS

Table 1 presents performance comparisons across the AIME24/25, LiveBench (reasoning), and GPQA Diamond benchmarks. Our proposed Once-More consistently outperforms all baselines across model sizes and task types. On mathematical reasoning tasks (AIME 2024/2025), Once-More achieves 3.4 to 10 point gains over raw generation, with the 14B model reaching 36.7% on AIME 2024 and 26.7% on AIME 2025. In contrast, Self-Refine and CRITIC show minimal improvements, gaining at most 2.1 to 4.5 points. For general reasoning (LiveBench), Once-More delivers 9.0 to 12.7 point improvements, while baselines achieve less than 1 point gain. On GPQA Diamond, all model sizes show consistent 3.6 to 7.6 point improvements with Once-More. Table 2 compares Once-More against supervised fine-tuning approaches, including MetaMath-SFT and S$^3$c-MATH. Once-More achieves the best performance on SVAMP across both model families (82.0% for Llama 3 8B, 89.0% for Qwen2 Math 7B) without any training. While S$^3$c-MATH outperforms Once-More on GSM8K for Llama (82.9% vs 79.0%), Once-More leads on Qwen2 Math (85.2% vs 84.8%). The results demonstrate that Once-More can match or exceed specialized training methods while maintaining model-agnostic deployment.

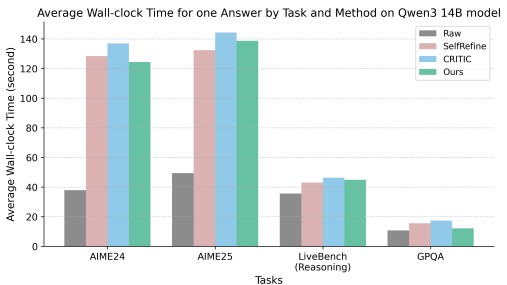

(a) Average wall-clock time for one answer.

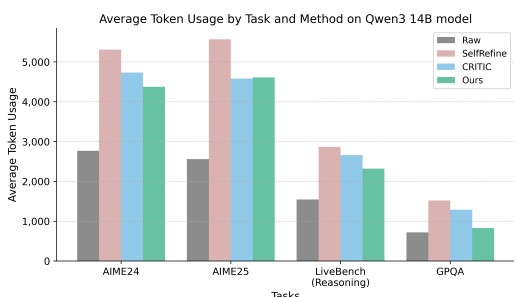

(b) Average token usage for one answer.

Figure 4: Run time comparisons across tasks on Qwen3-14B.

## 4.3 ANALYSIS AND ABLATION STUDY

**Scaling effects in self-correction.** On AIME 2024, Once-More's improvements scale progressively: a 25.6% relative gain for 4B, 36.4% for 8B, and 37.4% for 14B models. This scaling pattern suggests that self-correction benefits from richer internal representations in larger models. They can generate more informative feedback and possess more nuanced token distributions for effective logit redistribution. The same pattern holds across AIME 2025, where gains increase from 16.5% (4B) to 42% (14B). This indicates that Once-More can scale with model capacity, unlike post-hoc methods that show diminishing returns.

**Small models and the feedback quality problem.** Existing self-correction methods do not work on smaller models. Self-Refine shows no improvement on the 4B model for AIME 2024 and actually degrades performance on AIME 2025 for the 8B model (16.7%→15.5%). CRITIC fares slightly better but still achieves minimal gains. This is likely due to smaller models' limited capabilities to produce meaningful feedback on long outputs. Once-More mitigates this by combining verifier feedback with logit redistribution, ensuring corrections happen even when feedback quality is low. The finer unit-level granularity also makes feedback inherently easier for verifiers to provide.

**Mathematical reasoning improvement.** On AIME24 and AIME25, Once-More substantially outperforms CRITIC and Self-Refine. This is because mathematical errors are unforgiving; a single misstep can invalidate an entire solution. This makes existing post-hoc methods less reliable as they must locate the faulty step in a complete solution and regenerate a full answer. In contrast, Once-More's continuous monitoring catches mistakes as they occur, enabling correction at the point of failure rather than after completion.

**Efficiency analysis.** Figure 4 compares wall-clock time and token consumption across methods on Qwen3 14B. Despite its regeneration cycles, Once-More demonstrates competitive wall-clock time, matching or outperforming Self-Refine and CRITIC across all tasks while achieving superior accuracy. For token usage, Once-More consistently achieves the lowest consumption among self-correction methods, using 17-21% fewer tokens than Self-Refine on AIME tasks. This efficiency advantage persists across other benchmarks. Notably, GPQA Diamond exhibits shorter generations across all methods, providing fewer opportunities for continuous intervention and partially explaining the smaller performance gains on this task. Overall, Once-More achieves superior accuracy without incurring additional computational costs compared to existing self-correction approaches.

**Effects of verifier feedback and logit redistribution** Table 3 evaluates the contributions of feedback and logit redistribution on Qwen3-14B. Removing redistribution but keeping feedback yields 33.3% on AIME24 (+6.6) and 51.5% on GPQA (+3.5). Removing feedback but keeping redistribution gives 30.3% (+3.6) and 48.4% (+0.4). The full system reaches 36.7% and 55.6%, which are gains of +10.0 and +7.6 over raw. The effects are roughly additive on AIME24 and clearly more than additive on GPQA, suggesting that feedback and redistribution resolve different failure modes: feedback supplies semantic guidance about the error, and redistribution enforces exploration away from the previously chosen tokens so that the guidance actually changes the trajectory.

Table 3: Ablation study on Once-More components using Qwen3-14B. Both verifier feedback and logit redistribution contribute substantially, with the best accuracy performance from combining both.

| Method | AIME24 | GPQA |
|---|---|---|
| Raw | 26.7 | 48.0 |
| w/o Redistribution | 33.3 | 51.5 |
| w/o Feedback | 30.3 | 48.4 |
| Full Once-more | **36.7** | **55.6** |

Table 4: Unit length ablation study using Qwen3-14B, with mean accuracy (%) $\pm$ std. dev. over 3 runs.

| Benchmark | Unit length (sentences) | | | | | |
|---|---|---|---|---|---|---|
| | 1 | 2 | 4 | 32 | 64 | 128 |
| AIME24 | **36.6 $\pm$ 3.3** | 34.3 $\pm$ 1.9 | 35.5 $\pm$ 1.9 | 30.0 $\pm$ 2.7 | 26.7 $\pm$ 2.7 | 26.7 $\pm$ 2.7 |
| LiveBench | **52.3 $\pm$ 1.2** | 52.1 $\pm$ 1.0 | **52.3 $\pm$ 1.2** | 49.8 $\pm$ 1.7 | 50.3 $\pm$ 1.3 | 45.3 $\pm$ 0.6 |
| GPQA-D | 55.6 $\pm$ 1.8 | 55.5 $\pm$ 1.3 | **56.1 $\pm$ 1.5** | 49.7 $\pm$ 1.0 | 49.3 $\pm$ 0.3 | 50.0 $\pm$ 0.7 |

**Effects of answer unit granularity**   Table 4 evaluates the contributions of answer unit granularity on Qwen3-14B model under default hyper-parameter setting. The results do not indicate significant performance differences for small unit lengths ($\leq 4$ sentences). This shows that the proposed Once-More framework is robust under fine-grained segmentation. However, when the unit length increases significantly ($\geq 32$ sentences), performance drops across all three benchmarks and converges to the performance of the raw model (without Once-More) at a unit length of 128.

Table 5: Accuracy (%) results under different Producer–Verifier configurations and raw output.

| Producer: | Qwen-14B | | | | Qwen-4B | | | |
|---|---|---|---|---|---|---|---|---|
| Verifier: | 14B | 8B | 4B | Raw | 14B | 8B | 4B | Raw |
| AIME24 | 36.6 | 36.6 | 30.0 | 26.6 | 33.3 | 23.3 | 16.7 | 13.3 |
| LiveBench | 52.5 | 49.5 | 46.5 | 44.0 | 41.0 | 38.5 | 33.0 | 20.3 |
| GPQA-D | 55.6 | 51.5 | 50.0 | 48.0 | 54.0 | 52.5 | 47.5 | 43.9 |

**Asymmetric settings of Producer/Verifier**   Table 5 presents the results using both configurations of the strong-producer / weak-verifier and the weak-producer / strong-verifier. Experiments conducted under default hyperparameters. Results show that, with a strong Producer (Qwen-14B), performance degrades only mildly with weaker verifiers (4B/8B), remaining substantially better than the raw baseline. With a weak Producer (Qwen-4B), increasing verifier strength (using 14B) yields massive improvements across all benchmarks.

## 5   CONCLUSION

We presented Once-More, a model-agnostic, training-free framework for continuous self-correction during generation, which monitors uncertainty at the unit level, invokes verifier feedback when needed, and enforces exploration via perplexity-guided logit redistribution. This fine-grained, intervene-as-you-go design efficiently reduces error propagation and turns incremental improvements at each unit into stronger end-to-end generations. When tested on diverse benchmarks and model sizes, the proposed Once-More consistently outperforms post-hoc refinement baselines while remaining token-efficient, demonstrating that controllable, inference-time steering can yield reliable gains without additional training. Despite these gains, we acknowledge specific limitations: the framework may struggle to trace errors rooted deep in the generation history, potentially leading to regeneration loops; confident errors (false negatives) may occasionally bypass the perplexity trigger; and the system's performance ceiling remains influenced by verifier quality. Future work will focus on addressing these challenges through dynamic rollback mechanisms and adaptive thresholding.

## REPRODUCIBILITY STATEMENT

To ensure the reproducibility of our work, we provide comprehensive implementation details throughout the paper and supplementary materials. The Once-More framework's complete algorithmic description appears in Appendix A.1, with mathematical formulations for perplexity computation (Section 3.2) and logit redistribution mechanism (Section 3.3). Experimental setup details, including all hyperparameters ($K = 5$, $\alpha = 1.0$, $\beta = 1.0$), model specifications (Qwen3 4B/8B/14B), and benchmark descriptions are provided in Section 4.1. We also detail our implementation procedures for Self-Refine and CRITIC, including specific adaptations for mathematical tasks. All experiments use publicly available models and benchmarks (AIME 2024/2025, LiveBench, GPQA Diamond, SVAMP, GSM8K) with standard evaluation metrics. We report results averaged over three independent runs to account for stochastic variation. A GitHub repository will be used to host the complete implementation of Once-More after this work is published. It will include the Producer-Verifier architecture, perplexity monitoring system, logit redistribution algorithm, and scripts to reproduce all experimental results.

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

# A APPENDIX

## A.1 ONCE-MORE ALGORITHM

---

**Algorithm 1** Once-More Framework

---

**Require:** Producer LLM, Tokenizer, Verifier set $\mathcal{R}$, Goal $x$, role prompt, max tokens $T_{\max}$
 1: **Init:**
   Build chat prompt from (role prompt, $x$);
   Encode to token id: step_ids.
   Set past $\leftarrow \emptyset$, Output $\leftarrow \emptyset$, tokens $\leftarrow 0$. consecutive fails $\leftarrow 0$
   Initialize ckpt_stack $\leftarrow [(0, step\_ids)]$, supp_stack $\leftarrow [None]$, suppression $\leftarrow$ *None*.
 2: **for** $i = 1$ to $T_{\max}$ **do**
 3:   **Propose:** Generate sentence $s$ and update $(past, step\_ids, suppression)$.
 4:   **Monitor:**
     Compute reliability $\rightarrow (ok, \lambda)$.
     Update supp_stack with $(\lambda, t_{1:L}, \ell_{1:L})$.
     Update tokens $\leftarrow tokens + L$.
 5:   **if** not $ok$ and $i > 1$ **then**
 6:     **Verify:** Verifier judge $(accept, F) \leftarrow \text{Judge}(\mathcal{R}, x, Output, s)$.
 7:     **if** not $accept$ **and** consecutive fails $< 4$ **then**
 8:       **Rollback & Guide:**
         Restore last checkpoint $(k^{\star}, ids^{\star})$;
         set suppression from supp_stack;
         trim past to $k^{\star}$;
         reset step_ids $\leftarrow ids^{\star}$;
         pop stacks;
         append feedback $F$ to step_ids;
         consecutive fails += 1
         **continue**.
 9:     **end if**
10:   **end if**
11:   **Commit:**
     Append $s$ to Output;
     push new checkpoints to ckpt_stack and supp_stack;
     reset suppression $\leftarrow$ *None*.
     reset consecutive $\leftarrow 0$
12:   **if** last token of step_ids is eos **or** tokens $> T_{\max}$ **then**
13:     **break**
14:   **end if**
15: **end for**
16: **return** Output

---

## A.2 QUANTILE CALIBRATION FOR THE RELIABILITY THRESHOLD

We set the verification trigger by calibrating a *unit perplexity* threshold from a short, held-out run.

**Inputs and unit count.** For each calibration prompt $i = 1, \dots, D$, run the Producer once *without* verification and compute a unit-level perplexity $P_{\text{unit}}$ for every emitted unit (as in Section 3.2). If the $i$-th output contains $U_i$ units, it contributes $U_i$ scalar values. The total number of calibration units is

$$M = \sum_{i=1}^{D} U_i.$$

Collect all unit perplexities as $\{P^{(m)}\}_{m=1}^{M}$.

**Empirical quantile.** Form the empirical distribution by *sorting the values themselves* in ascending order (this sorting step ignores the original time order), yielding the order statistics

$$P_{(1)} \leq P_{(2)} \leq \cdots \leq P_{(M)}.$$

Fix a target verification rate $\eta \in (0, 1)$ so that roughly the largest $\eta$ fraction of units will be verified at run time. Set $q = 1 - \eta$ and compute

$$h = 1 + (M - 1)\, q, \qquad k = \lfloor h \rfloor, \qquad \gamma = h - k,$$

then define the calibrated threshold

$$P_{\text{th}} = \begin{cases} P_{(1)}, & h \leq 1, \\ P_{(M)}, & h \geq M, \\ (1 - \gamma)\, P_{(k)} + \gamma\, P_{(k+1)}, & \text{otherwise.} \end{cases}$$

At inference time, verify a provisional unit $Y$ whenever $P_{\text{unit}}(Y) > P_{\text{th}}$.

## A.3 MATHEMATICAL PROPERTIES OF THE FRAMEWORK

We now establish formal properties of the Once-More framework's perplexity computation and probability adjustment mechanism.

### A.3.1 PROPERTIES OF PERPLEXITY-BASED MONITORING

**Property 1 (Perplexity Lower Bound).** For any position $j$ and $K \geq 1$, the token-level perplexity satisfies $\text{PPL}_j^{(K)} \geq 1$.

*Proof.* By definition, $\text{PPL}_j^{(K)} = \exp(\text{ANLL}_j^{(K)})$ where

$$\text{ANLL}_j^{(K)} = \frac{1}{K} \sum_{i=1}^{K} (-\log q_j(v_{j,i})) \tag{10}$$

Since $q_j(v_{j,i}) \in (0, 1]$ for valid probabilities, we have $-\log q_j(v_{j,i}) \geq 0$. The lower bound of 1 is achieved when $K = 1$ and $q_j$ places all mass on a single token (i.e., $q_j(v_{j,1}) = 1$), giving $-\log(1) = 0$ and $\exp(0) = 1$.

**Property 2 (Sensitivity Control via K).** Larger values of $K$ increase the sensitivity of perplexity to distribution spread, while smaller values focus on top candidates.

*Justification.* When $K$ is small, $\text{PPL}_j^{(K)}$ reflects only the most probable tokens. If the top token has a high probability, perplexity remains low even if tail probabilities are spread. As $K$ increases, more of the distribution is captured, increasing sensitivity to uncertainty in lower-ranked alternatives. This creates a trade-off: larger $K$ detects subtle uncertainties but increases variance; smaller $K$ provides conservative, stable estimates.

A.3.2 PROPERTIES OF GUIDED REGENERATION

**Property 3 (Mass Balance).** Given suppression factors satisfying $0 \leq \alpha_j r_j(\text{target}_j) < 1$ and redistribution coefficient $\kappa_j$ as defined in Section 3.3, the probability mass removed from the target token exactly equals the mass added to alternatives before normalization.

*Proof.* See Appendix A.4 for the complete derivation of $\kappa_j$, which explicitly enforces this constraint.

**Property 4 (Valid Probability Distribution).** The adjusted distribution $\tilde{q}_j(\cdot)$ is a valid probability distribution (non-negative, sums to 1).

*Proof.* By construction:

1. **Non-negativity:** For the target token, $s_j(\text{target}_j) = 1 - \alpha_j r_j(\text{target}_j) > 0$ by the constraint $\alpha_j r_j(\text{target}_j) < 1$. For alternatives, $s_j(v) = 1 + \kappa_j r_j(v)^\beta > 0$ since $\kappa_j > 0$ (as long as alternatives exist) and $r_j(v) \geq 0$. Since $q_j^{(2)}(v) \geq 0$ for all $v$, we have $q_j^{(2)}(v)s_j(v) \geq 0$.

2. **Normalization:** The denominator $\sum_{u \in \mathcal{V}} q_j^{(2)}(u)s_j(u) > 0$ by non-negativity and the fact that $q_j^{(2)}$ is a valid distribution. The normalization explicitly ensures $\sum_{v \in \mathcal{V}} \tilde{q}_j(v) = 1$.

**Property 5 (Monotone Suppression).** For fixed $q_j^{(2)}$ and importance ratios $r_j$, the adjusted probability $\tilde{q}_j(\text{target}_j)$ strictly decreases with $\alpha_j$ within the admissible range.

*Proof.* From the definition:

$$\tilde{q}_j(\text{target}_j) = \frac{q_j^{(2)}(\text{target}_j)(1 - \alpha_j r_j(\text{target}_j))}{\sum_u q_j^{(2)}(u)s_j(u)} \tag{11}$$

Taking the derivative with respect to $\alpha_j$ (applying the quotient rule and using the fact that $\kappa_j$ depends on $\alpha_j$), one can show that:

$$\frac{\partial \tilde{q}_j(\text{target}_j)}{\partial \alpha_j} < 0 \tag{12}$$

The key insight is that increasing $\alpha_j$ directly reduces the target token's weight while increasing alternatives' weights, creating a double effect.

**Property 6 (Controlled Exploration via $\beta$).** The tempering exponent $\beta$ controls the distribution of redistributed mass:

- $\beta = 0$: Mass spreads uniformly across all alternatives
- $\beta = 1$: Mass allocates proportionally to importance ratios $r_j(v)$
- $\beta > 1$: Mass concentrates on tokens with the largest importance ratios

*Justification.* The boost factor for alternatives is $1 + \kappa_j r_j(v)^\beta$. When $\beta = 0$, this becomes $1 + \kappa_j$ (constant across all alternatives), distributing mass uniformly. As $\beta$ increases, $r_j(v)^\beta$ amplifies differences: tokens with $r_j(v) > 1$ (model now favours more) receive exponentially more boost, while tokens with $r_j(v) < 1$ receive less. This creates an exploration-exploitation trade-off controlled by $\beta$.

**Property 7 (Stability of Adjustment).** When $q_j^{(2)}$ remains close to $q_j^{(1)}$ and $\beta \leq 1$, the KL divergence between the adjusted and original distributions scales linearly with maximum suppression:

$$\text{KL}\big(\tilde{q}_j \,\|\, q_j^{(2)}\big) = O(\alpha_{\max}) \tag{13}$$

*Proof sketch.* The KL divergence is:

$$\text{KL}\big(\tilde{q}_j \,\|\, q_j^{(2)}\big) = \sum_v \tilde{q}_j(v) \log \frac{\tilde{q}_j(v)}{q_j^{(2)}(v)} \tag{14}$$

When $\alpha_j$ is small, $s_j(v) \approx 1$ for all $v$, making $\tilde{q}_j(v) \approx q_j^{(2)}(v)/Z$ where $Z \approx 1$. A Taylor expansion around $\alpha_j = 0$ shows that the leading term is linear in $\alpha_j$. The condition $\beta \leq 1$ ensures that the redistribution doesn't create sharp peaks that would increase divergence. For detailed calculation, note that:

$$\log \frac{\tilde{q}_j(v)}{q_j^{(2)}(v)} = \log s_j(v) - \log Z \approx s_j(v) - 1 - (Z - 1) \tag{15}$$

where both $s_j(v) - 1$ and $Z - 1$ are $O(\alpha_j)$.

**Property 8 (Alignment and Diffusion Regularization).** The combination of distance-decayed alignment and Gaussian diffusion prevents both over-localization and excessive spread of suppression signals.

*Justification.* The distance decay $\exp(-(|i - j|/\tau)^\gamma)$ reduces suppression for tokens matched far from their original position, preventing spurious long-range alignments from causing inappropriate suppression. The Gaussian diffusion with bandwidth $\sigma$ then smooths the resulting signal:

$$\bar{u}_j = \frac{1}{Z_j} \sum_{i=1}^{n} \exp\left(-\frac{(j - i)^2}{2\sigma^2}\right) \hat{u}_i^{\rightarrow} \tag{16}$$

As $\sigma \to 0$, this reduces to position-wise suppression $\bar{u}_j \to \hat{u}_j^{\rightarrow}$ (localized). As $\sigma$ increases, suppression spreads to neighboring positions (regularized). The final blend $u_j^\star = 0.5\hat{u}_j^{\rightarrow} + 0.5\bar{u}_j$ balances both effects, preventing over-fitting to specific positions while maintaining locality.

## A.4 Derivation of the Redistribution Constant

We now derive the formula for $\kappa_j$ that ensures probability mass conservation during the adjustment process.

**Setup.** At position $j$, we adjust the original distribution $q_j^{(2)}(v)$ by applying scaling factors $s_j(v)$:

$$s_j(v) = \begin{cases} 1 - \alpha_j r_j(v), & v = \text{target}_j \\ 1 + \kappa_j r_j(v)^\beta, & v \neq \text{target}_j \end{cases} \tag{17}$$

where:

- $\alpha_j \in (0, 1)$ is the suppression strength

- $r_j(v) = q_j^{(2)}(v)/\max(\varepsilon, \bar{q}_j^{(1)}(v))$ is the importance ratio

- $\beta \geq 0$ controls redistribution sharpness

- $\kappa_j$ is the redistribution coefficient to be determined

**Mass Conservation Constraint.** For a valid probability distribution, the total probability mass before and after adjustment must be equal. Define:

**Mass removed from target:**

$$\Delta^- = q_j^{(2)}(\text{target}_j) - q_j^{(2)}(\text{target}_j) \cdot s_j(\text{target}_j) \tag{18}$$

$$= q_j^{(2)}(\text{target}_j) - q_j^{(2)}(\text{target}_j) \cdot (1 - \alpha_j r_j(\text{target}_j)) \tag{19}$$

$$= \alpha_j r_j(\text{target}_j) \cdot q_j^{(2)}(\text{target}_j) \tag{20}$$

**Mass added to alternatives:**

$$\Delta^+ = \sum_{u \neq \text{target}_j} \left[ q_j^{(2)}(u) \cdot s_j(u) - q_j^{(2)}(u) \right] \tag{21}$$

$$= \sum_{u \neq \text{target}_j} \left[ q_j^{(2)}(u) \cdot (1 + \kappa_j r_j(u)^\beta) - q_j^{(2)}(u) \right] \tag{22}$$

$$= \sum_{u \neq \text{target}_j} q_j^{(2)}(u) \cdot \kappa_j r_j(u)^\beta \tag{23}$$

$$= \kappa_j \sum_{u \neq \text{target}_j} q_j^{(2)}(u) \cdot r_j(u)^\beta \tag{24}$$

**Solving for $\kappa_j$.** Setting $\Delta^- = \Delta^+$ for mass conservation:

$$\alpha_j r_j(\text{target}_j) \cdot q_j^{(2)}(\text{target}_j) = \kappa_j \sum_{u \neq \text{target}_j} q_j^{(2)}(u) \cdot r_j(u)^\beta \tag{25}$$

Solving for $\kappa_j$:

$$\boxed{\kappa_j = \frac{\alpha_j r_j(\text{target}_j) \cdot q_j^{(2)}(\text{target}_j)}{\sum_{u \neq \text{target}_j} q_j^{(2)}(u) \cdot r_j(u)^\beta}} \tag{26}$$

**Normalization.** After applying the scaling factors, we normalize to obtain the final distribution:

$$\tilde{q}_j(v) = \frac{q_j^{(2)}(v) \cdot s_j(v)}{\sum_{u \in \mathcal{V}} q_j^{(2)}(u) \cdot s_j(u)} \tag{27}$$

By construction (mass conservation), the numerator of the normalization constant equals:

$$\sum_u q_j^{(2)}(u) \cdot s_j(u) = q_j^{(2)}(\text{target}_j) \cdot s_j(\text{target}_j) + \sum_{u \neq \text{target}_j} q_j^{(2)}(u) \cdot s_j(u) \tag{28}$$

$$= \left[ q_j^{(2)}(\text{target}_j) - \Delta^- \right] + \left[ \sum_{u \neq \text{target}_j} q_j^{(2)}(u) + \Delta^+ \right] \tag{29}$$

$$= \sum_u q_j^{(2)}(u) + (\Delta^+ - \Delta^-) \tag{30}$$

$$= 1 + 0 = 1 \tag{31}$$

Thus, the distribution is properly normalized even before the explicit normalization step, confirming that our choice of $\kappa_j$ correctly preserves mass.

**Numerical Safeguards.** In practice, we implement several safeguards:

- Use $\varepsilon = 10^{-8}$ in importance ratios to avoid division by zero
- Clip $\alpha_j r_j(\text{target}_j)$ to ensure it remains strictly less than 1
- If the denominator in $\kappa_j$ is extremely small (no viable alternatives), either reduce $\beta$ toward 0 to spread mass broadly or cap $\kappa_j$ and trigger a fresh generation attempt
- Verify that $\sum_u q_j^{(2)}(u)s_j(u) > 0$ before normalization

## A.5 ABLATION STUDY

### A.5.1 ANALYSIS OF SUPPRESSION STRENGTH $\alpha$

The ablation study in $\alpha$ (suppression strength) is carried out using the Qwen-14B model on AIME24, Livebench, and GPQA-Diamond. Results are reported in Table 6 from a single run. All other parameters are chosen as the default value.

Table 6: Ablation study on suppression strength $\alpha$ using Qwen3 14B. Results from a single run.

|  | $\alpha = 0.1$ | $\alpha = 0.5$ | $\alpha = 1$ | $\alpha = 1.5$ | $\alpha = 2$ |
|---|---|---|---|---|---|
| AIME24 | 30.0 | 33.3 | 36.6 | 36.6 | 36.6 |
| LiveBench | 49.5 | 52.5 | 54.5 | 54.5 | 54.0 |
| GPQA-D | 50.5 | 51.5 | 52.0 | 52.5 | 51.0 |

For all benchmarks, performance is relatively stable for suppression strengths $\alpha \geq 1$. While a small suppression strength ($\alpha = 0.1$) dilutes the effect of our proposed Once-More Framework and achieves the worst performance in the ablation study.

### A.5.2 ANALYSIS OF DISTANCE DECAY FACTOR $\tau$

The ablation study on $\tau$ (distance decay factor) is conducted by the Qwen-14B model on AIME24, Livebench and GPQA-Diamond. The granularity of $\tau$ is set to 0.01 (extremely small $\tau$ representing decay to 0, suppression only works on the target token in this specific position), 0.5, 1, 1.5 and 100 (extremely large $\tau$ representing no decay, the target token will be equally suppressed regardless of its position). Results are reported in Table 7 from a single run. All other parameters are chosen as the default value.

Table 7: Ablation on distance decay factor $\tau$. Accuracy from a single run.

| Benchmark | $\tau = 0.01$ | $\tau = 0.5$ | $\tau = 1$ | $\tau = 1.5$ | $\tau = 100$ |
|---|---|---|---|---|---|
| AIME24 | 23.3 | 36.6 | 36.6 | 36.6 | 30.3 |
| LiveBench | 43.5 | 52.5 | 52.5 | 52.0 | 50.5 |
| GPQA-D | 45.5 | 54.5 | 55.0 | 55.0 | 52.5 |

The ablation results show a very interesting phenomenon. Without distance decay (an extremely small $\tau = 0.01$), model performance decreases significantly, becoming even worse than the raw model baseline. Under this setting, the Once-More framework only suppresses the target token if it is in the exact same position as in the previously rejected sequence. The model sometimes finds a trick to regenerate. Assuming this is the previously rejected unit:

```
This sequence contains an error and was therefore rejected
```

The producer first generates a few other words and then repeats exactly the same sequence as the previously rejected one, just like:

```
    Got it! This sequence contains an error and was therefore
↪   rejected
```

Since the position is mismatched by these beginning words, the later regenerated sequence is not suppressed at all. However, this will be subsequently rejected by the reviewer, causing the model to get stuck in a loop until the maximum token limit is reached.

Table 8: Ablation on diffusion bandwidth $\sigma$. Accuracy from a single run.

| Benchmark | $\sigma = 0.1$ | $\sigma = 1$ | $\sigma = 5$ | $\sigma = 10$ |
|---|---|---|---|---|
| AIME24 | 33.3 | 36.3 | 26.6 | 30.0 |
| LiveBench | 52.5 | 55.0 | 48.5 | 47.5 |
| GPQA-D | 50.5 | 52.5 | 49.0 | 49.5 |

### A.5.3 ANALYSIS OF DIFFUSION BANDWIDTH $\sigma$

Table 8 reports the effect of diffusion bandwidth $\sigma$, which controls how suppression spreads across neighbouring tokens. The results show that a large $\sigma(\geq 5)$ spreads the suppression too broadly. It dilutes the suppression effect on the target token and behaves similarly to using a very small suppression strength ($\alpha$). Conversely, a very small $\sigma$ restricted suppression only to the current token, making the adjustment too localized and also degrading accuracy.

### A.5.4 ANALYSIS OF PERPLEXITY THRESHOLD $\eta$

Since the parameter $K$ (top-$K$ for perplexity estimation) directly defines unit perplexity and consequently impacts the threshold ratio $\eta$, we conducted a joint ablation study of these two parameters. Table 9 and Table 10 show that, for small $K$ (1 or 5), changing $\eta$ primarily impacts runtime rather than accuracy. A looser threshold ($\eta = 50\%$) increases verification calls and computation time, while a stricter threshold ($\eta = 12.5\%$) improves efficiency. Large K values generally degrade accuracy.

Table 9: AIME24 results: accuracy and wall-clock time across perplexity threshold $\eta$ and top-$K$ settings.

|  | $\eta = 12.5\%$ | | $\eta = 25\%$ | | $\eta = 50\%$ | |
| --- | --- | --- | --- | --- | --- | --- |
|  | Acc. | Time (s) | Acc. | Time (s) | Acc. | Time (s) |
| $K = 1$ | 36.6 | 1693 | 36.6 | 3579 | 36.6 | 6448 |
| $K = 5$ | 36.6 | 2448 | 33.3 | 4133 | 36.6 | 6103 |
| $K = 10$ | 33.3 | 2397 | 33.3 | 3142 | 36.6 | 5954 |
| $K = 15$ | 30.0 | 2929 | 33.3 | 2851 | 33.3 | 7088 |

Table 10: GPQA results: accuracy and wall-clock time across perplexity threshold $\eta$ and top-$K$ settings.

|  | $\eta = 12.5\%$ | | $\eta = 25\%$ | | $\eta = 50\%$ | |
| --- | --- | --- | --- | --- | --- | --- |
|  | Acc. | Time (s) | Acc. | Time (s) | Acc. | Time (s) |
| $K = 1$ | 51.0 | 1987 | 54.5 | 3640 | 61.6 | 4953 |
| $K = 5$ | 47.5 | 2448 | 55.5 | 3848 | 55.0 | 4424 |
| $K = 10$ | 49.0 | 2397 | 49.5 | 2098 | 53.5 | 4244 |
| $K = 15$ | 48.4 | 2473 | 49.5 | 2486 | 55.0 | 3412 |

## A.6 ADDITIONAL OBSERVATIONS

### A.6.1 INTERMEDIATE REASONING BEHAVIOUR

Here is one example that the proposed Once-More framework exhibits intermediate reasoning behaviour. The following text is the partial output from AIME24 problem 83:

```
We are given a 2x3 grid where each cell contains a digit from 0 to
↪  9. We need to fill the grid such that:
   1. The **sum of the two numbers formed by reading left to right**
↪  (i.e., the top row and bottom row) is **999**.

   [Perplexity acceptable: False]
   [Judge] Output accepted.

   2. The **sum of the three numbers formed by reading top to bottom**
↪  (i.e., the first column, second column, and third column) is **99**.

   [Perplexity acceptable: True]
```

```
Let's denote the grid as follows:
$$
\begin{array}{|c|c|c|}
\hline
a & b & c \\
\hline
d & e & f \\
\hline
\end{array}
$$
### Step 1: Understand the horizontal sums
We are told that:
$$
abc + def = 999
$$
Here, $abc$ and $def$ are 3-digit numbers formed by the digits in
the top row and bottom row, respectively.

[Perplexity acceptable: False]
[Judge] Output accepted.

past_cache length: 400
That is:
- $abc = 100a + 10b + c$
- $def = 100d + 10e + f$
So:
$$
(100a + 10b + c) + (100d + 10e + f) = 999 \tag{1}
$$
### Step 2: Understand the vertical sums
We are told that:
$$
a + d + e = 99 \quad \text{(first column)} \\
b + e + f = 99 \quad \text{(second column)} \\
c + f = 99 \quad \text{(third column)}
$$

[Perplexity acceptable: False]
[Judge] Output rejected.
[Rolling back to the corresponding checkpoint]
[Feedback message to the model]:
 (The vertical sums must involve multi-digit numbers, not single
 digits. Re-express the vertical sums as $ad + de + ef = 99$, etc.,
 or clarify the problem statement.)

Clarifying: the problem says "the sum of the three numbers formed by
reading top to bottom is 99". That means:
- The first column forms a number: $a \times 100 + d \times 10 +
...$, but that would be too large.

[Perplexity acceptable: False]
[Judge] Output rejected.
[Rolling back to the corresponding checkpoint]
[Feedback message to the model]:
 (Clarify how the top-to-bottom numbers are formed (e.g.,
 single-digit numbers or multi-digit). The current interpretation is
 unclear.)

Let's assume that the top-to-bottom numbers are formed by reading
each of the three columns as a single-digit value (i.e., just the
digit in each column). Then the sum of these three digits is 99,
which is impossible since the maximum sum of three digits is
$9+9+9=27$. So this interpretation is invalid.
```

```
    [Perplexity acceptable: False]
    [Judge] Output accepted.

    Let's try a different interpretation: The **top-to-bottom** numbers
↪   are formed by concatenating the two digits in each column to form a
↪   two-digit number.

    [Perplexity acceptable: False]
    [Judge] Output accepted.

    For example, if the first column has digits $a$ and $b$, then the
↪   number is $10a + b$. Then we can add up the three such numbers for
↪   the three columns, and that total should be 99.

    So now we have:

    - The top row is a 3-digit number: $100a + 10b + c$
    - The bottom row is a 3-digit number: $100d + 10e + f$
    - The sum of these two numbers is **999**:
      $(100a + 10b + c) + (100d + 10e + f) = 999$ | (Equation 1)

    - The sum of the three column-wise two-digit numbers is **99**:
      $(10a + d) + (10b + e) + (10c + f) = 99$ | (Equation 2)

    We are to find the number of such digit arrangements that satisfy
↪   both equations.
```

This partial output shows how Once-More conducted one successful self-correction.

## A.7 PRACTICAL IMPLEMENTATION NOTES

**Length changes and alignment.** If the revised unit $Y' = [y'_1, \ldots, y'_m]$ has $m \neq n$:

- If $m < n$, positions $j > m$ do not apply; early stopping may be the correction.
- If $m > n$, positions $j > n$ are generated without suppression (they did not occur previously).
- For $j \leq \min(m, n)$ we apply the update above. Optionally, one can align by minimal edit distance when the new attempt diverges early; our experiments did not require this.

**Chunking strategy and context management**

- **Choosing the unit.** For prose, we default to sentence boundaries with a max token cap, splitting very long sentences into clauses; for code, we use syntax-aware spans (function, block, or diff hunk). Smaller units give finer correction. Larger units preserve coherence.
- **Boundary overlap.** We allow a short overlap window when proposing the next unit and let the Verifier inspect a sliding window that straddles the boundary to avoid seam artifacts.
- **Context growth.** For long outputs, we keep a rolling buffer of recent accepted text, optionally with compressed summaries of older content, to stay within context limits.
- **Multiple verifiers.** Factuality, format, and safety checks can run in parallel with conjunctive acceptance, or in stages (cheap checks first). Tool use follows standard agent patterns.

## A.8 EXAMPLE PROMPTS

Below are the prompts we used in Once-More for the *Producer*, *Verifier A* (Formal & Local), *Verifier B* (Global/Sanity), and an optional *Adjudicator*.

**Shared verifier `SYSTEM` (use for both verifiers).**

```
     You are a verifier. You judge exactly one CURRENT_SPAN in a partial
↪    solution.
     Do not solve the whole task. Be precise and conservative.

     Inputs:
     - TASK: the problem/question
     - ACCEPTED_CONTEXT: the already-accepted prior steps/state
     - CURRENT_SPAN: the producer's proposed new step(s) to add now
```

### GPQA-STYLE GRADUATE Q&A (SCIENCE)

#### Producer (`SYSTEM+USER`)

```
     [SYSTEM]
     You are a graduate-level problem solver. Solve the problem step by
↪    step, separated by a period. Your answer should be chosen from
↪    options A, B, C, D and end with:
     Final answer: <A/B/C/D>

     [USER]
     QUESTION:
     {{GPQA_QUESTION_STEM_AND_OPTIONS}}
```

#### Verifier A (Formal & Local) (`USER` only; use shared SYSTEM above)

```
     [USER]
     TASK: {{GPQA_QUESTION_STEM_AND_OPTIONS}}
     ACCEPTED_CONTEXT: {{PRIOR_NOTES_OR_NONE}}
     CURRENT_SPAN: {{PRODUCER_PARAGRAPH}}

     You are a very strict verifier with Rubric:
     R1 algebraic legality (no invalid cancellations, correct radical/log
↪    rules),
     R2 arithmetic accuracy,
     R3 domain/branch/constraints respected and stated,
     R4 check carefully for the hold of equations/inequalities,
     R5 check the scientific correctness of each claim.
     R6 Flag missing premises, leaps, or contradictions.
     R7 Counterfactual statement Checking.

     Does the input unit in the right track to achieve the goal, given
↪    the context? Verify the input unit with your role and task. Only
↪    reject the answer according to your role.
     The input may not necessarily solve the goal directly as there are
↪    more details in the upcoming text. If yes, answer Yes. If no, answer
↪    No and provide repair hints beginning with 'Suggestion' to improve
↪    in no more than 20 words.
```

#### Verifier B (Global/Sanity) (`USER` only; use shared SYSTEM above; optional)

```
     [USER]
     TASK: {{GPQA_QUESTION_STEM_AND_OPTIONS}}
```

```
ACCEPTED_CONTEXT: {{PRIOR_NOTES_OR_NONE}}
CURRENT_SPAN: {{PRODUCER_PARAGRAPH}}

Perform:
S1 quick alternative micro-derivation or spot-check,
S2 sanity bounds / physical plausibility,
S3 attempt 1-2 counterexamples,
S4 flag hidden assumptions.

Allowed error_tags:
↪  ["counterexample","bounds","assumption","consistency"].
```

### AIME-STYLE OLYMPIAD MATHEMATICS

#### Producer (`SYSTEM+USER`)

```
[SYSTEM]
You are an AIME problem solver. Produce one clean step-wise
↪  derivation and end with:
Final answer: \boxed{<integer 0-999>}

[USER]
PROBLEM:
{{AIME_PROBLEM_TEXT}}
```

#### Verifier A (Formal & Local) (`USER` only; use shared SYSTEM above)

```
[USER]
TASK: {{AIME_PROBLEM_TEXT}}
ACCEPTED_CONTEXT: {{PRIOR_ACCEPTED_STEPS_OR_NONE}}
CURRENT_SPAN: {{PRODUCER_STEPS}}

Rubric:
R1 algebraic legality (no invalid cancellations, correct radical/log
↪  rules),
R2 arithmetic accuracy,
R3 domain/branch/constraints respected and stated,
R4 local entailment from prior state,
R5 notation/variable hygiene,
R6 if final answer present: integer 0-999 and derived quantity
↪  matches.

Does the input unit in the right track to achieve the goal given the
↪  context? Verify the input unit with your role and task. Only reject
↪  the answer according to your role.
The input may not necessarily solve the goal directly as there are
↪  more details in the upcoming text. If yes, answer Yes. If no, answer
↪  No and provide repair hints beginning with 'Suggestion' to improve
↪  in no more than 20 words.
```

#### Verifier B (Global/Sanity) (`USER` only; use shared SYSTEM above; optional)

```
[USER]
TASK: {{AIME_PROBLEM_TEXT}}
ACCEPTED_CONTEXT: {{PRIOR_ACCEPTED_STEPS_OR_NONE}}
CURRENT_SPAN: {{PRODUCER_STEPS}}

Perform:
S1 quick recompute of the same claim via a different micro-tactic,
S2 sanity bounds/mod checks (e.g., parity, sign, rough magnitude),
```

```
      S3 try 1-2 simple counterexamples consistent with constraints,
      S4 flag hidden assumptions (e.g., x != 0).

      Allowed error_tags:
↪     ["counterexample","bounds","assumption","consistency"].
```

### LIVENBENCH (REASONING)

#### Producer (`SYSTEM+USER`)

```
      [SYSTEM]
      You are an problem solver. Solve the problem step by step seperated
↪     by period. This problem is guaranteed to have a solution. Should not
↪     exit without finding the exact solution.
      The final answer is wrapped as:\n<solution>...</solution>

      [USER]
      PROBLEM:
      {{LIVEBENCH_PROBLEM_TEXT}}
```

#### Verifier A (Formal & Local) (`USER` only; use shared SYSTEM above)

```
      [USER]
      TASK: {{LIVEBENCH_PROBLEM_TEXT}}
      ACCEPTED_CONTEXT: {{PRIOR_ACCEPTED_STEPS_OR_NONE}}
      CURRENT_SPAN: {{PRODUCER_STEPS}}

      Rubric (apply all):
      R1 factual correctness of claims vs. standard
↪     definitions/literature,
      R2 local logical entailment (no leaps),
      R3 unit/notation hygiene,
      R4 current answer matches reasoning,
      R5 the current answer is not necessarily complete.

      Does the input unit in the right track to achieve the goal given the
↪     context? Verify the input unit with your role and task. Only reject
↪     the answer according to your role.
      The input may not necessarily solve the goal directly as there are
↪     more details in the upcoming text.
      If yes, answer Yes. If no, answer No and provide repair hints
↪     beginning with 'Suggestion' that can be concatenated to the context
↪     to improve future answer.
      The suggestion should no more than 10 words.
```

### SVAMP

#### Producer (`SYSTEM+USER`)

```
      [SYSTEM]
      You are an problem solver. Solve the problem step by step seperated
↪     by period. This problem is guaranteed to have a solution. Should not
↪     exit without finding the exact solution.
      The final answer should be in the format: Final answer:
↪     \boxed{<integer 0-999>}

      [USER]
      PROBLEM:
      {{SVAMP_PROBLEM_TEXT}}
```

**Verifier A (Formal & Local) (`USER` only; use shared SYSTEM above)**

```
[USER]
TASK: {{SVAMP_PROBLEM_TEXT}}
ACCEPTED_CONTEXT: {{PRIOR_ACCEPTED_STEPS_OR_NONE}}
CURRENT_SPAN: {{PRODUCER_STEPS}}

R1 algebraic legality (no invalid cancellations, correct radical/log
↪ rules),
R2 arithmetic accuracy,
R3 check carefully for current formulating the real life problem
↪ into math symbols, be very sensitive to negative numbers
R4 check carefully for formulating the real life problem into math
↪ symbols,
R5 check carefully for any unresonable number during the
↪ calculation, which will not happen in real life situation
R6 Check carefully about whether the current answer is overthinking
↪ or too complex to be true in real life situation.
R7 Check carefully about whether the current context model the goal
↪ correctly

Verify the input unit with your role and task. Only reject the
↪ answer according to rubric. Do not reject for other reasons.
The input may not necessarily solve the goal directly as there are
↪ more details in the upcoming text. If yes, answer Yes. If no, answer
↪ No and provide repair hints beginning with 'Suggestion' to improve
↪ in no more than 20 words.
```

## GSM8K

**Producer (`SYSTEM+USER`)**

```
[SYSTEM]
You are an problem solver. Solve the problem step by step seperated
↪ by period. This problem is guaranteed to have a solution. Should not
↪ exit without finding the exact solution.
Final answer: \boxed{<integer 0-999>}

[USER]
PROBLEM:
{{GSM8K_PROBLEM_TEXT}}
```

**Verifier A (Formal & Local) (`USER` only; use shared SYSTEM above)**

```
[USER]
TASK: {{GSM8K_PROBLEM_TEXT}}
ACCEPTED_CONTEXT: {{PRIOR_ACCEPTED_STEPS_OR_NONE}}
CURRENT_SPAN: {{PRODUCER_STEPS}}

R1 algebraic legality (no invalid cancellations, correct radical/log
↪ rules),
R2 arithmetic accuracy,
R3 check carefully for current formulating the real life problem
↪ into math symbols, be very sensitive to negative numbers
R4 check carefully for formulating the real life problem into math
↪ symbols,
R5 check carefully for any unresonable number during the
↪ calculation, which will not happen in real life situation
R6 Check carefully about whether the current answer is overthinking
↪ or too complex to be true in real life situation.
```

```
    R7 Check carefully about whether the current context model the goal
↪   correctly

    Verify the input unit with your role and task. Only reject the
↪   answer according to rubric. Do not reject for other reasons.
    The input may not necessarily solve the goal directly as there are
↪   more details in the upcoming text. If yes, answer Yes. If no, answer
↪   No and provide repair hints beginning with 'Suggestion' to improve
↪   in no more than 20 words.
```

## A.9 USE OF LARGE LANGUAGE MODEL (LLM)

We used an LLM strictly for editorial assistance at the final drafting stage. Specifically, the LLM was used only to check for grammatical errors, fix typographical mistakes, and enhance sentence transitions for better readability. All technical content, including ideas, algorithms, proofs/derivations, experiments, analyses, tables/figures, and conclusions, was authored by the listed authors. The LLM did not generate, rewrite, or materially alter any scientific claims or results.

All LLM-suggested edits were manually reviewed and accepted or rejected by the authors. We retained authorship control at all times and ensured that no technical meaning was changed. The LLM is not an author and bears no responsibility for the paper's content. The authors assume full accountability for all claims and results.

