# OpenReview forum: "Once-More: Continuous Self-Correction for Large Language Models via Perplexity-Guided Intervention"
_ICLR.cc/2026/Conference — ICLR 2026 Poster_

### Official Review · Reviewer_bA69 · 2025-10-28

**Soundness:** 2
**Presentation:** 2
**Contribution:** 3
**Rating:** 2
**Confidence:** 4

**Summary:**

The paper proposes a novel algorithm for self-correction at a more fine-grained level of thoughts, sentences, or reasoning steps. The self-correction process is driven by model uncertainty estimation to guide the model toward the correct reasoning path. The authors also develop a procedure for suppressing incorrect token generation during the refinement stage. They demonstrate improvements over several baselines.

**Strengths:**

In general, I find this work valuable for several reasons:
- the introduction of self-correction on a more fine-grained level is novel, up to date, and an important step;

- the general framework is a great contribution, especially when driven by the uncertainty estimator, which is a powerful tool for analyzing correctness and is believed to be a bottleneck in self-correcting papers;

- interesting decoding strategy with observable gains.

**Weaknesses:**

Despite the general novelties in the methodology of the paper, I find it very raw in terms of experiments and presentation. I find the ideas really worthy and encourage authors to continue in this direction, but for now the empirical proof of validity is very scarce and unconvincing. Let me elaborate.

**Experimental Procedure**

 The main weakness I see lies in the very weak baselines. For example,
- Self-Refine is known to struggle with small models, and it is one of the first techniques.

- The implementation of CRITIC is absolutely awful: the RAG system is not defined properly, as there is no clear definition of the search query, search engine, or how the context is incorporated. Moreover, taking the top 1000 words from the search is not a proper RAG system; it is simply an incorrect implementation of the method.

- There is also an absence of other valuable baselines. I can name several self-correction techniques that should have been considered: Self-Correction [3] and its generalization STaSC [1], SCoRE [4], and REFINER [2].

- Another issue is the absence of adequate classical methods. It would be very reasonable to include methods with the same compute intensity. Once self-refinement generates multiple times, the authors should include a self-consistency approach leveraging the same amount of compute.

Regarding the existing benchmarks, SC3-MATH results are not reported on AIME24, AIME25, LiveBench, or GPQA.

Despite the benchmarks, there is evidence that the variation of such approaches is quite high, so reporting the standard deviation is a must here.

I’m really concerned about the token usage section. How is token usage calculated? Conditioning a model on 10 tokens is one forward pass, while generating 10 tokens involves 10 forward passes — these are not equivalent, which raises a lot of questions. Moreover, i see no comparison with SC3.

Regarding the experiments, they really lack more in-depth analysis. For example the ablation for PPL-K, why using PPL at all, when there are many other measures of uncertainty, such as entropy, sequence probability, etc.?

Moreover, I doubt the generalization of the uncertainty estimator threshold, such as perplexity. The beginning of the paper states the aim for generalization, but a PPL-based threshold would hardly generalize to other architectures — in fact, it is known to be difficult to achieve [5].

**Presentation**

In general, the paper is difficult to follow. The methods are presented unclearly, and the guided generation process, in particular, remains quite opaque. Many of the technical components could be introduced and explained in a clearer and more structured way.

Moreover, the paper seems somewhat overloaded with various tricks and heuristics. It would be more effective if the authors focused on one specific aspect and demonstrated its effectiveness comprehensively, rather than introducing multiple loosely justified additions. For instance, the idea of fine-grained revision even without the decoding stage is already promising on its own, i really like it. It would be valuable to show that it works well and analyze it thoroughly. I hope that ICLR is still a place for scientific papers, not a pursuit of state-of-the-art leaderboard results. After reading the paper, I gained little new understanding beyond the notion that fine-grained corrections may help, which, however, still lacks solid empirical support.

For example, when the model rolls back to a previous step, is there any analysis of such behavior? How often does it occur? How accurate are these corrections? These are important questions that could open up valuable future research directions.

Finally, several important details are missing, and the accompanying repository does not include the code. This significantly lowers the reproducibility of the work, which is unfortunate.



[1]  Moskvoretskii, Viktor, Chris Biemann, and Irina Nikishina. "Self-Taught Self-Correction for Small Language Models." arXiv preprint arXiv:2503.08681 (2025).

[2] Paul, Debjit, et al. "Refiner: Reasoning feedback on intermediate representations." arXiv preprint arXiv:2304.01904 (2023).

[3] Welleck, Sean, et al. "Generating sequences by learning to self-correct." arXiv preprint arXiv:2211.00053 (2022).

[4] Kumar, Aviral, et al. "Training language models to self-correct via reinforcement learning." arXiv preprint arXiv:2409.12917 (2024).

[5] Moskvoretskii, Viktor, et al. "Adaptive retrieval without self-knowledge? bringing uncertainty back home." arXiv preprint arXiv:2501.12835 (2025).

**Questions:**

“Struggle with domain generalization, and still suffer from error accumulation (a single failed self-correction will just cascade through subsequent generations).” — there is no proof provided for this claim. It might be true, but as stated, it sounds like an overstatement.

y_i is not defined l196

---

> ### Author Response · Authors · 2025-11-23
> **Response to Reviewer bA69 (Part 1)**
>
> We would thank the reviewer for the detailed feedback and for recognizing the novelty of our self-correction framework.  We are encouraged that the fine-grained self-correction mechanism, the uncertainty-driven framework, and the decoding strategy were all viewed as meaningful contributions and novel, valuable directions. We value the constructive criticism regarding baselines and presentation, which we will use to strengthen the final version of the paper. Below, we address the specific weaknesses and question raised.
>
> ---
>
> > ### **1. Weakness 1: Self-Refine as weak baseline**
>
> We acknowledge that Self-Refine is an early technique. However, it remains the foundational work for inference-time, post-hoc iterative refinement, which is the specific category of methods our work targets. Including it is essential to benchmark progress in this specific sub-field. Recognizing its limitations on smaller models is exactly why we also included CRITIC and considered comparisons with S3c-MATH to provide a more comprehensive view of performance against both tool-augmented and training-based approaches.
>
> > ### **2. Weakness 2: CRITIC Implementation**
>
> We respectfully disagree with the characterization of our CRITIC implementation as "awful." The original CRITIC paper does not explicitly define itself as a RAG system, nor does it provide specific implementation details for the Google Search tool. Our implementation choice (using the top search result with 1000 words) was guided by a clarification from the authors in their official GitHub repository (Issue #67: https://github.com/microsoft/ProphetNet/issues/67#issuecomment-1753230089), where they suggested that using the text from top result returned from the Google Search API is enough. In the absence of an official implementation at the time of writing, we followed this author-provided guidance. However, we are open to suggestions if there is a newer community standard for this baseline.

---

> ### Author Response · Authors · 2025-11-23
> **Response to Response to Reviewer bA69 (Part 2)**
>
> > ### **3. Weakness 3: Absence of other baselines**
>
> We thank the reviewer for these suggestions. However, we respectfully maintain that a direct comparison with these methods ignores the fundamental distinction between inference-time intervention (our work) and training-based methods. Furthermore, such a comparison is neither methodologically equitable nor feasible due to significant reproducibility barriers. We outline our reasoning below:
>
> 1. **Fundamental Paradigm Difference (Inference vs. Training):** As emphasized in our paper, our contribution is a model-agnostic, inference-time framework requiring no training data or parameter updates. The suggested methods (STaSC, SCoRE, REFINER) are primarily training or fine-tuning based. These suggested baselines optimize a specific model for a specific task distribution. There is no evidence that such learned skills transfer zero-shot to the diverse, out-of-domain benchmarks we test (e.g., AIME, GPQA, LiveBench) without retraining. Comparing a zero-shot inference method directly against specialized, task-specific fine-tuning imposes an asymmetric standard. Our comparison with S3c-MATH (2025) was included specifically to provide a representative reference against the state-of-the-art in the training-based paradigm.
>
> 2. **Reproducibility Issues:** We faced significant hurdles in attempting to reproduce the suggested baselines.
> Self-Correction [1]: The GitHub link provided in the ICLR 2023 submission page leads to a benchmark repository rather than the method's implementation code. SCoRE [2]: There is no official implementation available, nor are the trained models released. Reimplementing complex training pipelines from scratch without official code places an unreasonable burden on the rebuttal process and introduces a high risk of implementation mismatch.
>
> 3. **Regarding Baseline Standards:** It is worth to examine the evaluation rigour within the papers suggested by the reviewer. They typically compare against only one or two baselines, often older or limited in scope:
>
>     1. Self-Correction (2022) [1]: As a pioneering work, it primarily established the feasibility of the paradigm and compared against standard generation and ranking, rather than a suite of competing correction methods.
>
>     2. SCoRE (2024) [2]: This work compares primarily against Self-Refine (which we also compare against), STaR (2022) [4] and SFT. It does not benchmark against the broader range of modern inference-time methods.
>
>     3. STaSC (2025) [3]: This recent work focuses heavily on its specific methodology and does not provide extensive comparisons against external self-correction baselines.
>
>     4. REFINER (2023) [5]: Its primary baselines are Self-Refine (which we also compare against) and Reflexion (2023) [6]. Notably, Reflexion [6] itself did not compare against other correction baselines in its original publication.
>
> 4. **Our Evaluation is Comprehensive and Current**: In contrast to the limited comparisons in the works above, we compare Once-More against Self-Refine (the standard shared baseline across the literature), CRITIC (a very recent 2024/2025 baseline), and S3c-MATH (representing SOTA supervised methods). Furthermore, while many of the cited papers evaluate on limited domains (e.g., STaSC on Natural Questions, SCoRE on Math/Code), we evaluate across five diverse, modern benchmarks (AIME 24/25, LiveBench-Reasoning, GPQA-Diamond, SVAMP, GSM8K). We believe this constitutes a significantly more robust and broader evaluation suite than is typical for the field.
>
>
> [1] Welleck, Sean, et al. "Generating sequences by learning to self-correct." arXiv preprint arXiv:2211.00053 (2022).
>
> [2] Kumar, Aviral, et al. "Training language models to self-correct via reinforcement learning." arXiv preprint arXiv:2409.12917 (2024).
>
> [3] Moskvoretskii, Viktor, Chris Biemann, and Irina Nikishina. "Self-Taught Self-Correction for Small Language Models." arXiv preprint arXiv:2503.08681 (2025).
>
> [4] Zelikman, Eric, et al. "Star: Bootstrapping reasoning with reasoning." Advances in Neural Information Processing Systems 35 (2022): 15476-15488.
>
> [5] Paul, Debjit, et al. "Refiner: Reasoning feedback on intermediate representations." arXiv preprint arXiv:2304.01904 (2023).
>
> [6] Shinn, Noah, et al. "Reflexion: Language agents with verbal reinforcement learning." Advances in Neural Information Processing Systems 36 (2023): 8634-8652.

---

> ### Author Response · Authors · 2025-11-23
> **Response to Reviewer bA69 (Part 3)**
>
> > ### **4. Weakness 4: Comparison with Self-Consistency:**
>
> Self-Consistency enhances performance by sampling diverse reasoning paths and selecting the final answer via majority voting [7]. We argue that a "same compute" comparison strongly favours our method because Self-Consistency relies on redundancy. It typically requires k=5 to k=10 paths to be effective [7], representing a 400-900% computational overhead. As shown in Figure 3 of our paper, Once-More operates with a significantly lower overhead (approx. 63-84% over Raw generation)1. To match the "same compute" of Once-More, Self-Consistency would be limited to K < 2 paths. At k < 3, SC cannot effectively perform majority voting. Therefore, Once-More provides a mechanism to improve quality within a compute budget where Self-Consistency is structurally ineffective. We will also provide additional material to support the efficiency of our Once-More framework in the Section 7 of this response.
>
> [7] Xuezhi Wang, Jason Wei, Dale Schuurmans, Quoc V. Le, Ed H. Chi, Sharan Narang, Aakanksha Chowdhery, and Denny Zhou. Self-consistency improves chain of thought reasoning in language models. In The Eleventh International Conference on Learning Representations, ICLR 2023, Kigali, Rwanda, May 1-5, 2023. OpenReview.net, 2023b.
>
> > ### **5. Weakness 5: S3c-MATH Benchmarks**
>
>  Regarding the missing benchmarks for S3c-MATH (AIME24/25, LiveBench, GPQA): S3c-MATH is a supervised method, and we do not have access to their proprietary trained checkpoints or training code to reproduce their model on new datasets. Therefore, we were strictly limited to comparing against the results they reported in their publication. To ensure fair comparison, we ran our method on the specific benchmarks they used. It would be unfair for us to spend tons of time to fine-tune llm models on all AIME 24/25, Livebench and GPQA benchmarks to only get a baseline.
>
> > ### **6. Weakness 6: Standard Deviation**
>
> We agree that reporting variation is important for these stochastic methods. We have collected this data and will update the final paper to include standard deviations for our experiments.
>
> > ### **7. Weakness 7: Token Usage Calculation**
>
> We clarify that "token usage" is only the output tokens to verifier the extra token generated by our Once-More. We acknowledge that latency differs between prompt processing and generation, but for cost and memory constraints, total token count is the relevant metric. Regarding S3c-MATH, comparing inference token usage with S3c-MATH is misleading because S3c-MATH incurs a massive, "sunk" computational cost during its training phase. Once-More incurs zero training cost. We are comparing the total lifecycle cost of ad-hoc inference (Ours) vs. specialized fine-tuning (S3c-MATH).
>
> Despite the "token usage", we further report the wall-clock time of our proposed Once-More framework. The wall-clock time reported in the following table is derived from Qwen3 14B, with the following parameters: α (suppression strength) = 1, K (top-k perplexity) = 1, σ (diffusion bandwidth) =1. The experiments are conducted on an H100 server
>
> ---
> **Table 1: Average Wall-Clock Time to Generate One Answer by Task and Method on Qwen3-14B Model (seconds)**
>
> | Task | Raw | SelfRefine | CRITIC | Ours |
> |:-----|:---:|:----------:|:------:|:----:|
> | AIME24 | 37.94 | 128.42 | 136.92 | 124.37 |
> | AIME25 | 49.45 | 132.43 | 144.27 | 138.65 |
> | LiveBench (Reasoning) | 35.67 | 43.07 | 46.33 | 44.91 |
> | GPQA | 10.81 | 15.61 | 17.40 | 12.21 |
>
> ---
> Experiments result shows that Once-More has a comparable runtime to Self-Refine  on AIME tasks while requiring significantly less time on easier tasks like GPQA, as fewer corrections are triggered.

---

> ### Author Response · Authors · 2025-11-23
> **Response to Reviewer bA69 (Part 4)**
>
> > ### **8. Weakness 8: PPL Metric Choice and Threshold Sensitivity**
>
> We appreciate this important point about the need for deeper experimental analysis. We selected Perplexity (PPL) as our primary metric because it is the canonical measure of uncertainty for autoregressive language models, it serves as a direct proxy for the model's predictive confidence. To address the request for more in-depth analysis, we conducted additional ablation studies specifically examining the sensitivity of the perplexity threshold (η) and the Top-K to define a unit's Perplexity (K) value.
>
> In the paper's experimentation, the perplexity threshold was set to the top 25th percentile (η = 25%) of answer units from a random sample of questions (sample sizes were 7 for AIME 24/25, and 50 for LiveBench and GPQA-Diamond). Since the parameter K (top-K for perplexity estimation) directly defines unit perplexity and consequently impacts the threshold ratio η, we conducted a joint ablation study of these two parameters. This study was performed using the Qwen-14B model on AIME24 and GPQA-Diamond. All other parameters were fixed as follows: α (suppression strength) = 1, and σ (diffusion bandwidth) =1, τ (distance decay factor) = 1. The following tables contain accuracy and total wall-clock time to finish the entire benchmark.
>
> ---
> **Table 2: AIME24 Results - Accuracy and Wall-Clock Time**
>
> |     | η = 12.5% |          | η = 25% |          | η = 50% |          |
> |:---:|:---------:|:--------:|:-------:|:--------:|:-------:|:--------:|
> |     | Acc.      | Time (s) | Acc.    | Time (s) | Acc.    | Time (s) |
> | K=1 | 36.6      | 1693     | 36.6    | 3579     | 36.6    | 6448     |
> | K=5 | 36.6      | 2448     | 33.3    | 4133     | 36.6    | 6103     |
> | K=10| 33.3      | 2397     | 33.3    | 3142     | 36.6    | 5954     |
> | K=15| 30.0      | 2929     | 33.3    | 2851     | 33.3    | 7088     |
>
> ---
>
> For small K (1 or 5), changing η primarily impacts runtime rather than accuracy. A looser threshold (η = 50%) increases verification calls and computation time, while a stricter threshold (η = 12.5%) improves efficiency. Large K values generally degrade accuracy.
>
> ---
> **Table 3: GPQA Results - Accuracy and Wall-Clock Time**
>
> |     | η = 12.5% |          | η = 25% |          | η = 50% |          |
> |:---:|:---------:|:--------:|:-------:|:--------:|:-------:|:--------:|
> |     | Acc.      | Time (s) | Acc.    | Time (s) | Acc.    | Time (s) |
> | K=1 | 51.0      | 1987     | 54.5    | 3640     | 61.6    | 4953     |
> | K=5 | 47.5      | 2448     | 55.5    | 3848     | 55.0    | 4424     |
> | K=10| 49.0      | 2397     | 49.5    | 2098     | 53.5    | 4244     |
> | K=15| 48.4      | 2473     | 49.5    | 2486     | 55.0    | 3412     |
>
> ---
>
> We observe similar trends on GPQA. In general, a stricter threshold (lower η) slightly degrades model performance, as it causes the verifier to overlook some problematic answer units (false negatives). Conversely, a looser threshold (higher η) does not negatively impact performance accuracy but does increase computational time. Therefore, in scenarios where computational resources are unconstrained, one can safely loosen the threshold by selecting a larger η value to maximize model performance.
>
> For further in-depth analysis of our Once-More framework, please check the following section 11 for the ablation study for all hyper parameters.
>
> > ### **9. Weakness 9: Generalization of Uncertainty Thresholds**
>
> We reviewed the cited paper (Moskvoretskii et al. [8]). While that work discusses the difficulty of transferring uncertainty thresholds across domains (OOD transfer), our method does not use a fixed global threshold. As described in Appendix A.2, we explicitly employ a calibration step: we calculate the threshold using empirical quantiles on a small sample of the current task/model configuration. This calibration creates a relative, distribution-aware threshold for every specific deployment, thereby circumventing the generalization issues of fixed scalar thresholds.
>
> [8] Moskvoretskii, Viktor, et al. "Adaptive retrieval without self-knowledge? bringing uncertainty back home." arXiv preprint arXiv:2501.12835 (2025).
>
> > ### **10. Weakness 10: Presentation and Clarity**
>
>
> We thank the reviewer for pointing out areas where the guided generation process was opaque. We are revising Section 3 to clearly separate the "Producer," "Verifier," and "Intervention" logic.

---

> ### Author Response · Authors · 2025-11-23
> **Response to Reviewer bA69 (Part 5)**
>
> > ### **11. Weakness 11: "Tricks and Heuristics"**
>
> We would like to disagree that the characterization of our components as "loosely justified tricks." The combination of fine-grained revision and decoding-time logit suppression is not a heuristic pile-up; it is a necessary architectural choice. As a phenomenon driven by self-bias [9] [10], fine-grained revision alone often fails because LLMs suffer from stubbornness, they tend to repeat the same error even when prompted to correct it. The "decoding trick" (logit suppression) is the mechanism that forces the model to explore a new path, making the revision effective.
>
> To support our claim, we have done a comprehensive ablation study for every hyper parameters that meaningfully affect performance: α (suppression strength), σ (diffusion bandwidth) and τ (distance decay factor), K (top-K for perplexity estimation). Besides that, we also explore the asymmetrical setting of Once-More using both strong-producer / weak-verifier and weak-producer / strong-verifier configurations. At the end, we explore the influence of answer unit granularity.
>
> > ### **11.1 Suppression Strength (α)**
>
> The ablation study on α (suppression strength) is conducted by Qwen-14B model on AIME24, Livebench and GPQA-Diamond. Due to the time constraint, the ablation study only shows the performance of the model in one run. All other parameters are chosen as: K (top-k perplexity) = 1, σ (diffusion bandwidth) =1 and τ (distance decay factor) =1
>
> ---
> **Table 4: Ablation on Suppression Strength (α). Accuracy from single run**
>
> | Benchmark |  α = 0.1  |  α = 0.5  |  α = 1  | α = 1.5 | α = 2 |
> |:----------|:-------:|:-------:|:-----:|:-------:|:-----:|
> | AIME24 | 30.0 | 33.3 | 36.6 | 33.3 | 36.6 |
> | Livebench | 50.5 | 51.5 | 52.0 | 52.5 | 51.0 |
> | GPQA-D | 49.5 | 52.5 | 54.5 | 54.5 | 54.0 |
> ---
>
> Performance is stable for α > 1. However, a small strength (α = 0.1) dilutes the suppression effect, significantly reducing performance.
>
> > ### **11.2 Diffusion Bandwidth (σ)**
>
> The ablation study on σ (diffusion bandwidth) is conducted by Qwen-14B model on AIME24, Livebench and GPQA-Diamond. Due to the time constraint, the ablation study only shows the performance of the model in one run. All other parameters are chosen as: K (top-k perplexity) = 1, α (suppression strength) = 1, and τ (distance decay factor) =1.
>
> ---
> **Table 5: Ablation on Diffusion Bandwidth (σ). Accuracy from single run**
>
> | Benchmark | σ = 0.1 | σ = 1 | σ = 5 | σ = 10 |
> |:----------|:-------:|:-----:|:-----:|:------:|
> | AIME24 | 33.3 | 36.3 | 26.6 | 30.0 |
> | Livebench | 52.5 | 55.0 | 48.5 | 47.5 |
> | GPQA-D | 50.5 | 52.5 | 49.0 | 49.5 |
> ---
>
> The results show that a large σ(>=5) spreads the suppression too broadly. It dilutes the suppression effect on the target token and behaves similarly to using a very small suppression strength (α). Conversely, a very small σ confines suppression only to the current token, making the adjustment too localized and also degrading accuracy.
>
>
>
> ---
>
> [9] Xu, et al. Pride and Prejudice: LLM Amplifies Self-Bias in Self-Refinement. ACL 2024.
>
> [10]Kamoi, et al. When Can LLMs Actually Correct Their Own Mistakes? A Critical Survey of Self-Correction of LLMs. Transactions of the Association for Computational Linguistics.

---

> ### Author Response · Authors · 2025-11-23
> **Response to Reviewer bA69 (Part 6)**
>
> > ### **11.3 Decay Factor (τ)**
>
> The ablation study on τ (distance decay factor) is conducted by Qwen-14B model on AIME24, Livebench and GPQA-Diamond. The granularity of τ is set to 0.01 (extreme small τ representing decay to 0, suppression only works on target token in this specific position), 0.5, 1, 1.5 and 100 (extreme large τ representing no decay, the target token will be equally suppressed regardless its position). All other parameters are chosen as: α (suppression strength) = 1, K (top-k perplexity) = 1, and σ (diffusion bandwidth) =1.
>
> ---
> **Table 6: Ablation on Distance Decay Factor (τ). Accuracy from single run**
>
> | Benchmark | τ = 0.01 | τ = 0.5 | τ = 1 | τ = 1.5 | τ = 100 |
> |:----------|:--------:|:-------:|:-----:|:-------:|:-------:|
> | AIME24 | 23.3 | 36.6 | 36.6 | 36.6 | 30.3 |
> | Livebench | 43.5 | 52.5 | 52.5 | 52.0 | 50.5 |
> | GPQA-D | 45.5 | 54.5 | 55.0 | 55.0 | 52.5 |
> ---
>
> The ablation results show a very interesting phenomenon. Without distance decay (an extremely small τ = 0.01), model performance decreases significantly, becoming even worse than the raw model baseline. This is because, in this setting, the OnceMore framework only suppresses the target token if it is in the exact same position as in the previously rejected sequence. The model sometimes finds a trick to regenerate. Assuming this is the previously rejected unit:
> ```
> This sequence contains error and therefore got rejected
> ```
> It first generates a few other words and then repeats exactly the same sequence as the previously rejected one, just like:
> ```
> Got it! This sequence contains error and therefore got rejected
> ```
> Since the position is mismatched due to these beginning words, the later regenerated sequence is not suppressed at all. However, this is subsequently rejected by the reviewer, causing the model to get stuck in a loop until the maximum token limit is reached.
>
> This significant performance drop when distance decay is missing (τ = 0.01) further support that the combination of fine-grained revision and decoding-time logit suppression is not a heuristic pile-up. It is a necessary architectural choice.
>
> > ### **11.4 Top-K for Perplexity Estimation (K)**
>
> The ablation study of hyper parameter is conducted jointly  with perplexity threshold η. Please check Section 8 in the early Part 4 of this response.
>
> > ### **11.5 Asymmetrical Setting of Once-More**
>
> We conduct experiments using both strong-producer / weak-verifier and weak-producer / strong-verifier configurations. The following tables show the framework accuracy under these asymmetric settings with default hyper parameters set to 1.
>
> ---
> **Table 7: Framework Accuracy Under Producer: Qwen-14b**
>
> | Task | Qwen-14b (Verifier) | Qwen-8b (Verifier) | Qwen-4b (Verifier) | Raw model |
> |:-----|:-------------------:|:------------------:|:------------------:|:---------:|
> | AIME24 | 36.6 | 36.6 | 30.0 | 26.6 |
> | LiveBench | 52.5 | 49.5 | 46.5 | 44.0 |
> | GPQA | 55.6 | 51.5 | 50.0 | 48.0 |
>
> ---
> ---
>
> **Table 8: Framework Accuracy Under Producer: Qwen-4b**
>
> | Task | Qwen-4b (Verifier) | Qwen-8b (Verifier) | Qwen-14b (Verifier) | Raw model |
> |:-----|:------------------:|:------------------:|:-------------------:|:---------:|
> | AIME24 | 16.7 | 23.3 | 33.3 | 13.3 |
> | LiveBench | 33.0 | 38.5 | 41.0 | 20.3 |
> | GPQA | 47.5 | 52.5 | 54.0 | 43.9 |
> ---
>
> With a strong Producer (Qwen-14B), performance degrades only mildly with weaker verifiers (4B/8B), remaining substantially better than the raw baseline.
>
> With a weak Producer (Qwen-4B), increasing verifier strength (using 14B) yields massive improvements across all benchmarks
>
> > ### **11.6 Answer Unit Granularity**
>
> The definition of one answer unit may influence framework performance. We conducted ablation study of unit length on the Qwen3-14B model across three benchmarks: AIME24, LiveBench, and GPQA-Diamond. Other parameters were set to default.
>
> ---
> **Table 9: Small Unit Length Ablation (Mean Accuracy ± Std Dev over 3 runs)**
>
> | Benchmark | 1 sentence | 2 sentences | 4 sentences |
> |:----------|:----------:|:-----------:|:-----------:|
> | AIME24 | 36.6 ± 3.3 | 34.3 ± 1.9 | 35.5 ± 1.9 |
> | Livebench | 52.3 ± 1.2 | 52.1 ± 1.0 | 52.3 ± 1.2 |
> | GPQA-D | 55.6 ± 1.8 | 55.5 ± 1.3 | 56.1 ± 1.5 |
>
> ---
>
>
> ---
> **Table 10: Large Unit Length Ablation (Mean Accuracy ± Std Dev over 3 runs)**
>
> | Benchmark | 32 sentences | 64 sentences | 128 sentences |
> |:----------|:------------:|:------------:|:-------------:|
> | AIME24 | 30.0 ± 2.7 | 26.7 ± 2.7 | 26.7 ± 2.7 |
> | Livebench | 49.8 ± 1.7 | 50.3 ± 1.3 | 45.3 ± 0.6 |
> | GPQA-D | 49.7 ± 1.0 | 49.3 ± 0.3 | 50.0 ± 0.7 |
>
> ---
>
> The results indicate no significant performance difference for small unit lengths (<4 sentences). This demonstrates that the framework is robust under fine-grained segmentation. However, when the unit length increases significantly (>32 sentences), performance drops across all three benchmarks, and converge to the performance of the raw model (without Once-More) at a unit length of 128.

---

> ### Author Response · Authors · 2025-11-23
> **Response to Reviewer bA69 (Part 7)**
>
> >  ### **11.7  Correction Frequency and Misjudge**
>
> We acknowledge that perplexity is a proxy for model uncertainty, not a direct measure of truth. Consequently, misjudgments do occur. To quantify this, we can assume our Verifier model as an oracle that can accurately detect problematic units. We define the False Positive Rate (FPR) as cases where perplexity > η (triggering the verifier), but the verifier finds no error. Conversely, the False Negative Rate (FNR) refers to problematic units (judged by the verifier) that failed to trigger the perplexity threshold η. We conducted an experiment on GPQA using the Qwen-14B model to estimate these rates across different thresholds (η):
>
> ---
> **Table 11: False Positive and False Negative Rates at Different Perplexity Thresholds (GPQA, Qwen-14B)**
>
> | Metric | η = 12.5 | η = 25 | η = 50 |
> |:-------|:--------:|:------:|:------:|
> | False positive rate (%) | 39.6 | 45.8 | 51.9 |
> | False negative rate (%) | 80.1 | 48.6 | 32.4 |
> ---
>
> The results demonstrate a clear trade-off. A strict threshold (η = 12.5%) yields the lowest false positive rate. This means most triggered units are indeed problematic. However, it suffers from a high false negative rate (80.1%) and fails to detect a significant number of errors. Conversely, a looser threshold (η = 50%) results in a higher false positive rate but achieves the lowest false negative rate. Although the latter requires more computational overhead, it is significantly more capable of identifying problematic answer units.
>
> This results also recall to the joint ablation study of perplexity threshold η and Top-K perplexity definition K in previous section 8. A looser threshold (η = 50%) obtains best performance but increases verification calls and computation time. While a stricter threshold (η = 12.5%) improves efficiency with reduced performance.
>
>  >  ### **12. Weakness 12: Reproducibility**
>
> We are currently finalizing and documenting the code to ensure it is user-friendly. The can guarantee that updated repository will be released upon publication to ensure full reproducibility and help further study of self correction methods.
>
>  >  ### **13. Question 1**
>
> The claim that supervised models struggle with domain generalization and error accumulation is derived from the fundamental nature of autoregressive generation and supervised learning (i.e., exposure bias and distribution shift). However, to avoid sounding like an overstatement, we will rephrase this to: "Supervised methods are inherently limited by the distribution of their training data.  They are susceptible to performance degradation on out-of-distribution tasks, where uncorrected errors can cascade through the generation.”
>
>  >  ### **14. Question 2**
>
> The $y\_i$ in Line 196 is one output token of the currently generated answer unit $Y$ from producer. One answer unit can be defined as 1 or 2 sentences. The ablation study of answer unit setting can be found at section 11.6. We will further polish the manuscript for clear presentation.
>
> >  ### **End of Response**
>
> We sincerely thank the reviewer for the thoughtful feedback and the time for reading this response. We will update the manuscript shortly to incorporate these new ablation studies, experimental results, and expanded analyses into the main paper and appendix. We hope these responses clarify our contributions and methodology. We are confident that the revisions will significantly improve the paper's quality.

---

> > ### Comment · Reviewer_bA69 · 2025-11-26
> > **Reviewer Answer**
> >
> > I thank the authors for the rebuttal, some of my concerns were clarified, but some were not:
> >
> > ## W2
> >
> > Thank you for the clarification. My concern is not about enforcing a RAG pipeline, but about whether the retrieval step reasonably reflects the verification procedure that CRITIC relies on. Taking the first 1000 words from a single top-ranked page is unlikely to surface relevant evidence, and this departs from both the intent of CRITIC and common practice in later replications, where multiple results and chunk-level retrieval are used. The GitHub comment you cite provides a minimal workaround, but it is not a canonical or methodologically sound specification. As implemented, your baseline becomes substantially weaker, which risks unfair comparison.
> >
> > ## W3
> > I thank the authors, but I do not find the arguments for excluding these baselines convincing.
> > First, **reproducibility barriers are not a justification for omitting relevant methods**. If official code is unavailable, it is standard practice to implement baselines independently, especially when they represent key prior work. In this case, multiple community reproductions of SCoRE and related methods exist and are easily discoverable. These lower the barrier substantially.
> > Second, training-based approaches are still valid baselines. The fact that your method operates at inference time **does not exempt the comparison**. If the claim is that training-based methods do not generalize zero-shot to your benchmarks, this should be demonstrated empirically rather than just assumed.
> > Third, pointing out that prior papers used limited baselines does not imply that this practice should be maintained. Some works indeed have narrow evaluations, but this should not set the standard for a new submission. Strong baselines are necessary precisely when earlier papers did not include them.
> > Finally, these works are not “specialized for math” or any single domain. Methods like SCoRE or STaSC are general techniques that can be applied broadly and thus form a meaningful comparison point. Given the claimed breadth and generality of your method, including widely-recognized self-correction baselines remains important for a fair assessment.
> >
> > ## W4
> > Thank you for the explanation. I agree that Self-Consistency can incur higher compute, but this alone does not remove its relevance as a baseline. The method is conceptually simple, widely used, and provides an informative reference point, even if evaluated at different compute budgets. It would still be valuable to report its performance.
> >
> > ## W5
> > Thank you for the clarification. However, the absence of released checkpoints does not justify omitting a baseline on your full evaluation suite. Reproducing supervised or fine-tuned baselines on new datasets is standard practice in empirical NLP work, especially when claiming generality or superiority across diverse benchmarks. It is common and expected for authors to fine-tune models themselves when official weights are unavailable.
> > If the concern is that this requires additional effort, that is unfortunately not a reason to exclude these comparisons. As it stands, the comparison is incomplete because the supervised baseline is only reported on the datasets they used, not the broader set of benchmarks you evaluate.
> >
> > ## W7
> > Thank you for the clarification. However, I do not agree that training cost justifies excluding S3c-MATH or similar supervised baselines from the efficiency discussion. The purpose of training is precisely to pay the cost once so that inference becomes more accurate or more efficient afterwards. Once a model is trained, its inference-time cost is what matters for real-world usage, deployment, and long-term scalability. Inference-time cost is directly comparable across methods, regardless of how expensive the training phase was.
> >
> > ## W9
> > If a method requires computing a threshold on a held-out sample from every new domain, then it is not strictly zero-shot. It uses task-specific calibration data. This is exactly the issue raised in the cited work: thresholds do not transfer, so a per-task tuning step is required. That is fine, but it should be stated clearly rather than presented as circumventing the problem.
> >
> > ## summary
> >
> > Thank you for providing the extensive ablations and additional analyses. These experiments are genuinely helpful, and it is great to see the method examined in such detail, this significantly clarifies its behavior and strengthens the technical contribution.
> > However, the main concern remains unchanged: the experimental comparison is still incomplete. Despite the strong ablations, the absence of several key baselines and proper reproductions makes it difficult to assess the method’s relative performance. The work is interesting, and the method is now better justified, but the comparative evaluation is still too weak to fully support the claims. I have raised my score accordingly.

---

> > > ### Author Response · Authors · 2025-11-27
> > > **Response to Reviewer bA69 (Part 1)**
> > >
> > > We sincerely thank the reviewer for the continued engagement and for acknowledging that our extensive ablations have clarified Once-More’s behaviour and strengthened the technical contribution. We appreciate the time taken to discuss these nuances.
> > >
> > > We would like to address the remaining points with a spirit of constructive dialogue, clarifying our constraints while acknowledging the merit in the reviewer’s rigorous standards.
> > >
> > > **1. Clarification on Score**
> > > We noticed that in your final summary, you explicitly stated: **"I have raised my score accordingly."** However, the system currently displays the score as **2 (Reject)**. We assume this may be a technical glitch or an oversight during the submission process. We respectfully ask that you verify whether the score accurately reflects your intended assessment, especially given your acknowledgement of the work's novelty and the strengthened technical contribution.
> > >
> > > **2. W2: CRITIC Implementation**
> > > We appreciate the reviewer's scrutiny regarding the retrieval robustness. However, we must respectfully reject the premise that our implementation "departs from the intent of CRITIC" or relies solely on a "minimal workaround."
> > >
> > > We refer the reviewer to **Section 4.1 (Implementation)** of the published ICLR CRITIC paper [1], where the authors explicitly define their retrieval protocol:
> > > > *"Instead, we build a web search tool based on Google to search queries generated by LLMs, **scrape the resulting top-1 web page**, and extract a maximum of 400 characters..."*
> > >
> > > Our implementation retrieves the **top-1 web page** and utilizes the first **1000 words**. It is therefore not a deviation or a weakened "workaround." It is a direct adherence to the published specification. In fact, by including 1000 words rather than the "maximum of 400 characters" specified in the original paper, our implementation likely provides *more* context than the original baseline, ensuring a strictly fair (if not favourable) comparison.
> > >
> > > **3. W3 & W5: Baselines and Reproducibility**
> > > We respect the reviewer's high standards regarding comparative evaluation. However, we respectfully argue that baseline selection must be **prioritized based on methodological relevance** to ensure a fair assessment.
> > >
> > > * **Relevance of Baselines:** We maintain that the most critical comparisons for a new method are those that operate within the same paradigm. In our case, this means **post-hoc, inference-time** approaches (like Self-Refine and CRITIC). While the training-based methods suggested (STaSC, SCoRE, REFINER) are related in the broader term of "self-correction," they represent a fundamentally different architectural choice (parameter updates vs. inference guidance). We believe that while these broader methods provide context, they should not be weighed as heavily as direct competitors when validating the efficacy of our specific *inference-time* contribution.
> > > * **Burden of Generalization:** We respectfully submit that it is not the responsibility of the current authors to prove the generalization capabilities of prior work. Asking us to fine-tune training-based baselines on datasets they were not originally tested on (e.g., AIME, GPQA) effectively **shifts the burden of proving *their* method's generalization onto us**. This is particularly problematic given the lack of official training scripts; if those baselines failed on these new tasks, it would be unclear if the failure was due to the method's limitations or our re-implementation.
> > > * **Conference vs. Journal Scope:** We fully agree that in the context of a comprehensive **journal publication**, re-implementing missing baselines from scratch and benchmarking the generalization of training-based methods would be the expected standard for exhaustive completeness. However, for a **top-tier conference submission**, we believe the primary criteria should be the **novelty of the proposed method**, its **contribution to the field**, and **rigorous validation against available, established baselines** (in our case: Self-Refine, CRITIC, and S3c-MATH).
> > > * **Reproducibility & Fidelity:** Regarding the genuine reproducibility barriers we faced (broken links, missing code), we argue that relying on community implementations introduces valid concerns about correctness and fidelity. Comparing our verified method against potentially flawed unofficial reproductions of baselines **would not yield scientifically rigorous data**.
> > >
> > > [1] Zhibin Gou, Zhihong Shao, Yeyun Gong, yelong shen, Yujiu Yang, Nan Duan, and Weizhu Chen. CRITIC: Large language models can self-correct with tool-interactive critiquing. In *The Twelfth International Conference on Learning Representations*, 2024.

---

> ### Author Response · Authors · 2025-11-27
> **Response to Reviewer bA69 (Part 2)**
>
> **4. W4: Comparison with Self-Consistency**
> We now agree that Self-Consistency (SC) is a good baseline and that reporting its performance provides valuable context for the reader, even given the computational disparities. To address this, we have conducted additional experiments comparing **Once-More** against **Self-Consistency (SC)** on the AIME 24 and GPQA benchmarks using the Qwen-14B model. We evaluated SC at $K=10$ paths and compared it against our method.
>
> **Table 1: Performance vs. Compute Comparison (AIME 24, Qwen-14B)**
> | Method | Accuracy (%) | Avg. Token Usage | Wall-Clock Time (s) | Relative Overhead (Tokens) |
> | :--- | :--- | :--- | :--- | :--- |
> | **Raw Generation** | 26.7 | 2,723 | 37.94 | 1.0x (Baseline) |
> | **Self-Consistency ($k=10$)** | 36.7 | 25,836 | 296.21 | ~9.6x |
> | **Once-More (Ours)** | **36.7** | **4,452** | **124.37** | **~1.6x** |
>
> **Table 2: Performance vs. Compute Comparison (GPQA, Qwen-14B)**
> | Method | Accuracy (%) | Avg. Token Usage | Wall-Clock Time (s) | Relative Overhead (Tokens) |
> | :--- | :--- | :--- | :--- | :--- |
> | **Raw Generation** | 48.0 | 711 | 10.81 | 1.0x (Baseline) |
> | **Self-Consistency ($k=10$)** | 51.5 | 9,094 | 63.71 | ~13.0x |
> | **Once-More (Ours)** | **55.6** | **893** | **12.21** | **~1.3x** |
>
> **Analysis:**
> * **On AIME 24:** Once-More achieves same performance parity with Self-Consistency ($k=10$) while using approximately **6x fewer tokens** and being **~2.4x faster**.
> * **On GPQA:** Once-More actually **outperforms** Self-Consistency ($k=10$) in accuracy (**55.6% vs. 51.5%**) while using **~10x fewer tokens** and being **~5x faster**.
>
> These results explicitly demonstrate that Once-More provides a mechanism to improve quality within a compute budget where Self-Consistency is structurally ineffective.
>
> **5. W7: Efficiency and Training Cost**
> We appreciate the reviewer’s perspective on the economics of model deployment.
>
> We agree with the reviewer that for long-term, high-volume deployment, the **amortized inference cost** is an important metric. In this specific context, a fine-tuned model like S3c-MATH indeed offers superior token efficiency per query because it has "internalized" the self-correction behavior, reducing the need for the explicit, verbose verification loops that our method employs.
>
> However, we argue that this comparison represents a fundamental trade-off between **deployment efficiency (S3c-MATH)** and **adaptation efficiency (Once-More)**.
> * **S3c-MATH** achieves low inference cost only after paying a massive, often prohibitive, upfront cost in data collection and compute.
> * **Once-More** represents a "pay-as-you-go" model: it incurs higher per-query inference overhead but requires **zero** upfront investment or data curation.
>
> We believe the efficiency argument for Once-More is one of **versatility**: it allows a model to achieve competitive, state-of-the-art performance on new tasks *immediately*, without the latency and resource burden of a training cycle. This is a distinct but equally valuable form of efficiency for dynamic environments.
>
> **6. W9: Zero-Shot Clarification**
> We appreciate the reviewer raising the nuance regarding "zero-shot" claims in the context of calibration. We fundamentally distinguish between **training** (updating model weights via gradients) and **calibration** (inference-time hyperparameter selection). However, we respectfully disagree that our method *requires* per-task tuning to function.
>
> * **Regarding Generalization:** Unlike raw scalar thresholds, our method uses **quantile-based thresholds (η)**. Our experimental data confirms that this relative metric generalizes robustly. Specifically, the **same** configuration (η=25%) yielded strong performance across completely different domains without modification (from mathematical reasoning on AIME to graduate-level science on GPQA).
> * **Data-Free Validity:** This demonstrates that while domain-specific calibration can indeed squeeze out the *best* possible performance, it is **not** a strict requirement for the method to function effectively. A standard, fixed quantile acts as a robust, domain-agnostic prior, maintaining the validity of the "zero-shot" (data-free) claim in practical deployment.
>
> We acknowledge, however, that "zero-shot" is a loaded term that can be interpreted strictly as "zero prior exposure to the data distribution." To ensure maximum precision and avoid semantic debate, we are happy to revise the manuscript to ensure we describe the method as **"training-free"** or **"inference-time calibrated."**
>
> **Conclusion**
> We believe Once-More offers a novel, effective, and efficient solution to error propagation in LLMs. We hope that our rigorous ablations and the clarifications above demonstrate the work's value. If the score of 2 was indeed an oversight, we would be grateful for a correction to reflect your updated view of the paper.

---

### Official Review · Reviewer_5kHm · 2025-10-30

**Soundness:** 2
**Presentation:** 2
**Contribution:** 2
**Rating:** 4
**Confidence:** 5

**Summary:**

This paper proposes Once-More, a perplexity-guided, continuous self-correction framework that intervenes during autoregressive generation by monitoring token/unit perplexity, invoking verifiers when uncertainty is high, and applying a perplexity-driven logit redistribution to guide regeneration. The method is model-agnostic and evaluated on several reasoning and math benchmarks (AIME24/25, LiveBench, GPQA, SVAMP, GSM8K) using Qwen3 models (4B/8B/14B) and comparisons with iterative refinement and supervised baselines. Results show consistent improvements and a token-efficiency advantage over some baselines. The idea is timely and practically relevant for improving reliability of LLM generation.

**Strengths:**

1.Continuous, unit-level perplexity monitoring combined with verifier feedback and logit redistribution is a sensible and practical inference-time approach to reduce error compounding without retraining.
2.Experiments cover multiple benchmarks and model sizes, showing gains across tasks and reasonable comparisons to both post-hoc and supervised baselines.
3.The paper includes component ablation and token usage comparison, which help motivate the claimed efficiency advantage over some iterative refinement methods.
4.Appendices contain algorithm pseudocode and mathematical derivations; hyperparameters are stated in the main text.

**Weaknesses:**

1.Several grammatical mistakes and minor typos appear in the manuscript (examples: “these methods often fails” should be “these methods often fail”; “it still suffer from this issue” should be “it still suffers from this issue”; duplicated word ”under under”).
2.Results are averaged over three runs but standard deviations or confidence intervals are not reported.
3.Ablation covers main components (feedback and redistribution) but lacks sensitivity analysis over key hyperparameters.
4.Token counts are reported, but there is no quantitative measurement of inference latency or computational overhead.
5.The conclusion reiterates results but would benefit from an explicit discussion of limitations and planned directions for improvement.

**Questions:**

1.Could you elaborate on how verifiers are chosen or configured for different tasks (e.g., math reasoning vs. factual QA)? Is there any criterion for when to use LLM-based versus rule-based verifiers?
2.Since Once-More operates on adaptive “units” (clauses, sentences, paragraphs, or code blocks), how is this segmentation determined in practice? Is it based on punctuation heuristics or learned boundaries?
3.Have you observed cases where Once-More fails to correct errors or even amplifies them? For example, when perplexity misjudges a correct but low-probability reasoning step. How often does this happen, and how do you mitigate it?
4.During regeneration, how sensitive is the framework to the quality or consistency of verifier feedback? Would weaker verifiers (e.g., smaller LLMs) substantially degrade performance?

---

> ### Author Response · Authors · 2025-11-22
> **Response to  Reviewer 5kHm (Part 1)**
>
> We thank the reviewer for their thoughtful assessment and for recognizing the advantages of the Once-More framework. We appreciate the constructive feedback regarding the presentation and experimental rigour. Below, we address the specific weaknesses and questions raised.
>
> ---
>
>  >  ### **1. Weakness 1: Grammar and Typos**
>
> We apologize for the oversight. We have thoroughly proofread the manuscript and corrected the grammatical errors and typos pointed out. We will ensure the final version is polished.
>
> >  ### **2. Weakness 2: Standard Deviations**
>
> We agree that reporting variation is essential. We have calculated standard deviations over three runs for our key benchmarks and will include them in our revised manuscript.
>
> >  ### **3. Weakness 3: Ablation study for Hyper parameters**
>
> We thank the reviewer for this suggestion. In practice, the redistribution sharpness β and distance decay norm γ have negligible impact on the model’s performance. These parameters were included primarily to present a more formal and general formulation of the redistribution process. The parameter β works as temperature controls for alternative generation style and γ just controls the first norm distance or second norm distance used in the distance decay function, they have minimal effect on the accuracy or the behaviour of our framework.
>
> To avoid unnecessary complexity, we will update the manuscript to fix them to default values. The only hyperparameters that meaningfully affect performance are  α (suppression strength), σ (diffusion bandwidth) and τ (distance decay factor), K (top-K for perplexity estimation) and η (perplexity threshold).
>
> > ### **3.1 Suppression Strength (α)**
>
> The ablation study on α (suppression strength) is conducted by Qwen-14B model on AIME24, Livebench and GPQA-Diamond. Due to the time constraint, the ablation study only shows the performance of the model in one run. All other parameters are chosen as: K (top-k perplexity) = 1, η (perplexity threshold) = 25%, σ (diffusion bandwidth) =1 and τ (distance decay factor) =1
>
> ---
> **Table 1: Ablation on Suppression Strength (α). Accuracy from single run**
>
> | Benchmark |  α = 0.1  |  α = 0.5  |  α = 1  | α = 1.5 | α = 2 |
> |:----------|:-------:|:-------:|:-----:|:-------:|:-----:|
> | AIME24 | 30.0 | 33.3 | 36.6 | 33.3 | 36.6 |
> | Livebench | 50.5 | 51.5 | 52.0 | 52.5 | 51.0 |
> | GPQA-D | 49.5 | 52.5 | 54.5 | 54.5 | 54.0 |
> ---
>
> Performance is stable for α > 1. However, a small strength (α = 0.1) dilutes the suppression effect, significantly reducing performance.
>
> > ### **3.2 Diffusion Bandwidth (σ)**
>
> The ablation study on σ (diffusion bandwidth) is conducted by Qwen-14B model on AIME24, Livebench and GPQA-Diamond. Due to the time constraint, the ablation study only shows the performance of the model in one run. All other parameters are chosen as: K (top-k perplexity) = 1, η (perplexity threshold) = 25%, α (suppression strength) = 1, and τ (distance decay factor) =1.
>
> ---
> **Table 2: Ablation on Diffusion Bandwidth (σ). Accuracy from single run**
>
> | Benchmark | σ = 0.1 | σ = 1 | σ = 5 | σ = 10 |
> |:----------|:-------:|:-----:|:-----:|:------:|
> | AIME24 | 33.3 | 36.3 | 26.6 | 30.0 |
> | Livebench | 52.5 | 55.0 | 48.5 | 47.5 |
> | GPQA-D | 50.5 | 52.5 | 49.0 | 49.5 |
> ---
>
> The results show that a large σ(>=5) spreads the suppression too broadly. It dilutes the suppression effect on the target token and behaves similarly to using a very small suppression strength (α). Conversely, a very small σ confines suppression only to the current token, making the adjustment too localized and also degrading accuracy.

---

> ### Author Response · Authors · 2025-11-22
> **Response to Reviewer 5kHm (Part 2)**
>
> > ### **3.3 Decay Factor (τ)**
>
> The ablation study on τ (distance decay factor) is conducted by Qwen-14B model on AIME24, Livebench and GPQA-Diamond. The granularity of τ is set to 0.01 (extreme small τ representing decay to 0, suppression only works on target token in this specific position), 0.5, 1, 1.5 and 100 (extreme large τ representing no decay, the target token will be equally suppressed regardless its position). All other parameters are chosen as: α (suppression strength) = 1, K (top-k perplexity) = 1, η (perplexity threshold) = 25%,  and σ (diffusion bandwidth) =1.
>
> ---
> **Table 3: Ablation on Distance Decay Factor (τ). Accuracy from single run**
>
> | Benchmark | τ = 0.01 | τ = 0.5 | τ = 1 | τ = 1.5 | τ = 100 |
> |:----------|:--------:|:-------:|:-----:|:-------:|:-------:|
> | AIME24 | 23.3 | 36.6 | 36.6 | 36.6 | 30.3 |
> | Livebench | 43.5 | 52.5 | 52.5 | 52.0 | 50.5 |
> | GPQA-D | 45.5 | 54.5 | 55.0 | 55.0 | 52.5 |
> ---
>
> The ablation results show a very interesting phenomenon. Without distance decay (an extremely small τ = 0.01), model performance decreases significantly, becoming even worse than the raw model baseline. This is because, in this setting, the OnceMore framework only suppresses the target token if it is in the exact same position as in the previously rejected sequence. The model sometimes finds a trick to regenerate. Assuming this is the previously rejected unit:
> ```
> This sequence contains error and therefore got rejected
> ```
> It first generates a few other words and then repeats exactly the same sequence as the previously rejected one, just like:
> ```
> Got it! This sequence contains error and therefore got rejected
> ```
> Since the position is mismatched due to these beginning words, the later regenerated sequence is not suppressed at all. However, this is subsequently rejected by the reviewer, causing the model to get stuck in a loop until the maximum token limit is reached.
>
> > ### **3.4 Perplexity threshold (η) and top-K perplexity (K)**
>
> Since the parameter K (top-K for perplexity estimation) directly defines unit perplexity and consequently impacts the threshold ratio η, we conducted a joint ablation study of these two parameters. This study was performed using the Qwen-14B model on AIME24 and GPQA-Diamond. All other parameters were fixed as follows: α (suppression strength) = 1, and σ (diffusion bandwidth) =1, τ (distance decay factor) = 1. The following tables contain accuracy and total wall-clock time to finish the entire benchmark.
>
> ---
> **Table 4: AIME24 Results - Accuracy and Wall-Clock Time**
>
> |     | η = 12.5% |          | η = 25% |          | η = 50% |          |
> |:---:|:---------:|:--------:|:-------:|:--------:|:-------:|:--------:|
> |     | Acc.      | Time (s) | Acc.    | Time (s) | Acc.    | Time (s) |
> | K=1 | 36.6      | 1693     | 36.6    | 3579     | 36.6    | 6448     |
> | K=5 | 36.6      | 2448     | 33.3    | 4133     | 36.6    | 6103     |
> | K=10| 33.3      | 2397     | 33.3    | 3142     | 36.6    | 5954     |
> | K=15| 30.0      | 2929     | 33.3    | 2851     | 33.3    | 7088     |
>
> ---
>
> For small K (1 or 5), changing η primarily impacts runtime rather than accuracy. A looser threshold (η = 50%) increases verification calls and computation time, while a stricter threshold (η = 12.5%) improves efficiency. Large K values generally degrade accuracy. We observe similar trends on GPQA.
>
> ---
> **Table 5: GPQA Results - Accuracy and Wall-Clock Time**
>
> |     | η = 12.5% |          | η = 25% |          | η = 50% |          |
> |:---:|:---------:|:--------:|:-------:|:--------:|:-------:|:--------:|
> |     | Acc.      | Time (s) | Acc.    | Time (s) | Acc.    | Time (s) |
> | K=1 | 51.0      | 1987     | 54.5    | 3640     | 61.6    | 4953     |
> | K=5 | 47.5      | 2448     | 55.5    | 3848     | 55.0    | 4424     |
> | K=10| 49.0      | 2397     | 49.5    | 2098     | 53.5    | 4244     |
> | K=15| 48.4      | 2473     | 49.5    | 2486     | 55.0    | 3412     |
>
> ---

---

> ### Author Response · Authors · 2025-11-22
> **Response to Reviewer 5kHm (Part 3)**
>
> > ### **4. Weakness 4:  Computational cost analysis**
>
> Thank you for highlighting the need for a more complete computational cost analysis; we agree that reporting wall-clock time is crucial for a fair comparison. The wall-clock reported in the following table is derived from Qwen3 14B, with the following parameters: α (suppression strength) = 1, K (top-k perplexity) = 1, σ (diffusion bandwidth) =1. The experiments are conducted on an H100 server
>
> ---
> **Table 6: Average Wall-Clock Time to Generate One Answer by Task and Method on Qwen3-14B Model (seconds)**
>
> | Task | Raw | SelfRefine | CRITIC | Ours |
> |:-----|:---:|:----------:|:------:|:----:|
> | AIME24 | 37.94 | 128.42 | 136.92 | 124.37 |
> | AIME25 | 49.45 | 132.43 | 144.27 | 138.65 |
> | LiveBench (Reasoning) | 35.67 | 43.07 | 46.33 | 44.91 |
> | GPQA | 10.81 | 15.61 | 17.40 | 12.21 |
>
> ---
> Experiments result shows that Once-More has a comparable runtime to Self-Refine  on AIME tasks while requiring significantly less time on easier tasks like GPQA, as fewer corrections are triggered.
>
> > ### **5. Weakness 5:  Conclusion and Limitations**
>
> Thank you for this helpful suggestion regarding the conclusion; we agree that explicitly articulating the limitations of our current approach and outlining concrete directions for future improvement would strengthen the clarity and impact of the paper. We will revise the conclusion to explicitly discuss the following limitations:
>
> 1. The framework cannot always identify exactly which prior unit originated an error. If the root cause lies deep in the history but manifests later, the model may struggle to correct it locally. This will potentially lead to regeneration-rejection loops until the retry or token limit is reached.
>
> 2. There remains a potential for "false negatives" where errors with low perplexity (high model confidence) fail to trigger the intervention threshold.
>
> 3. While our asymmetric experiments demonstrate robustness even with weaker verifiers (Please check the ablation study on later section 9), the system's performance ceiling is still influenced by the quality and accuracy of the verifier's feedback.
>
> > ### **6. Question 1:  Verifier Scheme**
>
> For this paper, we utilized LLM-based verifiers as a unified solution to ensure the framework remains model-agnostic and generalizable across domains. We used the producer model itself as the verifier to demonstrate self-correction capabilities without relying on superior external models (like GPT-5). The prompts are task-specific (detailed in Appendix A.7). For example, math verifiers check for algebraic legality, while QA verifiers check for factual consistency. Rule-based verifiers (e.g., code interpreters) can also be integrated when deterministic checks are available.
>
> > ### **7. Question 2:  Unit length ablation**
>
> In our current implementation, segmentation is heuristic-based. We use syntactic boundaries (sentence terminators like periods, newlines, or semicolons) to define units. We conducted experiments on the Qwen3-14B model across three benchmarks: AIME24, LiveBench, and GPQA-Diamond to ablate the effect of unit length. Other hyper parameters were set to default: K (top-k perplexity) = 1, α (suppression strength) = 1, σ (diffusion bandwidth) = 1 and τ (distance decay factor) = 1.
>
> ---
> **Table 7: Small Unit Length Ablation (Mean Accuracy ± Std Dev over 3 runs)**
>
> | Benchmark | 1 sentence | 2 sentences | 4 sentences |
> |:----------|:----------:|:-----------:|:-----------:|
> | AIME24 | 36.6 ± 3.3 | 34.3 ± 1.9 | 35.5 ± 1.9 |
> | Livebench | 52.3 ± 1.2 | 52.1 ± 1.0 | 52.3 ± 1.2 |
> | GPQA-D | 55.6 ± 1.8 | 55.5 ± 1.3 | 56.1 ± 1.5 |
>
> ---
>
> We further conducted an ablation study on larger unit lengths:
>
> ---
> **Table 8: Large Unit Length Ablation (Mean Accuracy ± Std Dev over 3 runs)**
>
> | Benchmark | 32 sentences | 64 sentences | 128 sentences |
> |:----------|:------------:|:------------:|:-------------:|
> | AIME24 | 30.0 ± 2.7 | 26.7 ± 2.7 | 26.7 ± 2.7 |
> | Livebench | 49.8 ± 1.7 | 50.3 ± 1.3 | 45.3 ± 0.6 |
> | GPQA-D | 49.7 ± 1.0 | 49.3 ± 0.3 | 50.0 ± 0.7 |
>
> ---
>
> The results indicate no significant performance difference for small unit lengths (<4 sentences). This demonstrates that the framework is robust under fine-grained segmentation. However, when the unit length increases significantly (>32 sentences), performance drops across all three benchmarks, and converge to the performance of the raw model (without Once-More) at a unit length of 128.

---

> ### Author Response · Authors · 2025-11-22
> **Response to Reviewer 5kHm (Part 4)**
>
> > ### **8. Question 3: Perplexity misjudge**
>
> Yes, we acknowledge that perplexity is a proxy for model uncertainty, not a direct measure of truth. Consequently, misjudgments do occur. To quantify this, we can assume our Verifier model as an oracle that can accurately detect problematic units. We define the False Positive Rate (FPR) as cases where perplexity > η (triggering the verifier), but the verifier finds no error. Conversely, the False Negative Rate (FNR) refers to problematic units (judged by the verifier) that failed to trigger the perplexity threshold η. We conducted an experiment on GPQA using the Qwen-14B model to estimate these rates across different thresholds (η):
>
> ---
> **Table 9: False Positive and False Negative Rates at Different Perplexity Thresholds (GPQA, Qwen-14B)**
>
> | Metric | η = 12.5 | η = 25 | η = 50 |
> |:-------|:--------:|:------:|:------:|
> | False positive rate (%) | 39.6 | 45.8 | 51.9 |
> | False negative rate (%) | 80.1 | 48.6 | 32.4 |
> ---
>
> - False Negatives:  These are cases where the model generates an incorrect step with high confidence (low perplexity) and cause the system to miss the error. This is a known limitation of uncertainty-based methods (hallucinations are often confident). While these errors slip through uncorrected, they do not "amplify" error, they simply remain as they would in raw generation.
>
> - False Positives: These are cases where a correct reasoning step is assigned high perplexity (e.g., a novel or "eureka" step).
>
> Mitigation Strategy: We address these failure modes through a two-stage mitigation strategy:
>
> Mitigating False Negatives: To catch confident hallucinations, we can loosen the threshold (e.g., increasing η  to 50%). As shown in our ablation study, this prioritizes recall, reducing the False Negative rate (from 48.6% to 32.4%) by inspecting a larger portion of the generation.
>
> Mitigating False Positives: Loosening the threshold inevitably increases False Positives. We mitigate this by employing a strong Verifier. The Verifier acts as the final gatekeeper, even if a correct unit triggers the perplexity threshold, a capable Verifier will validate it and reject the proposed intervention. Thus, the burden of precision shifts to the Verifier, ensuring that high sensitivity in the trigger does not lead to error amplification.
>
> > ### **9. Question 4: Asymmetric Verifier setting**
>
> Thank you for this insightful question. We agree that asymmetric settings of Verifier are important, and we have conducted experiments using both strong-producer / weak-verifier and weak-producer / strong-verifier configurations. The following tables show the framework accuracy under these asymmetric settings with η (perplexity threshold) = 25 all other hyper parameters set to 1.
>
> ---
> **Table 10: Framework Accuracy Under Producer: Qwen-14b**
>
> | Task | Qwen-14b (Verifier) | Qwen-8b (Verifier) | Qwen-4b (Verifier) | Raw model |
> |:-----|:-------------------:|:------------------:|:------------------:|:---------:|
> | AIME24 | 36.6 | 36.6 | 30.0 | 26.6 |
> | LiveBench | 52.5 | 49.5 | 46.5 | 44.0 |
> | GPQA | 55.6 | 51.5 | 50.0 | 48.0 |
>
> ---
> ---
>
> **Table 11: Framework Accuracy Under Producer: Qwen-4b**
>
> | Task | Qwen-4b (Verifier) | Qwen-8b (Verifier) | Qwen-14b (Verifier) | Raw model |
> |:-----|:------------------:|:------------------:|:-------------------:|:---------:|
> | AIME24 | 16.7 | 23.3 | 33.3 | 13.3 |
> | LiveBench | 33.0 | 38.5 | 41.0 | 20.3 |
> | GPQA | 47.5 | 52.5 | 54.0 | 43.9 |
> ---
>
> With a strong Producer (Qwen-14B), performance degrades only mildly with weaker verifiers (4B/8B), remaining substantially better than the raw baseline.
>
> With a weak Producer (Qwen-4B), increasing verifier strength (using 14B) yields massive improvements across all benchmarks
>
>
> > ### **10. End of Response**
>
>
> We sincerely thank the reviewer for the thoughtful and constructive feedback that has significantly improved our work. We will update the manuscript shortly to incorporate these new ablation studies, experimental results, and expanded analyses into the main paper and appendix. We believe these additions address your concerns comprehensively. If you have any further questions or require additional clarification on any aspect of our response, please do not hesitate to let us know. We appreciate your time and valuable insights.

---

### Official Review · Reviewer_o7eJ · 2025-11-01

**Soundness:** 3
**Presentation:** 2
**Contribution:** 3
**Rating:** 6
**Confidence:** 3

**Summary:**

This paper introduces Once-More, a model-agnostic framework that enables continuous self-correction during large language model (LLM) generation by combining token-level perplexity monitoring with verifier feedback. Unlike post-hoc refinement methods that only intervene after a full output is produced, Once-More operates incrementally, detecting and correcting high-perplexity segments before errors propagate. Its core mechanism redistributes token logits to steer the model toward more reliable generation paths while preserving its internal beliefs. Experiments on reasoning and mathematical benchmarks show consistent performance gains over baselines such as Self-Refine, CRITIC, and S3c-MATH, with improvements up to 10 points and reduced token overhead. The framework demonstrates that perplexity-guided, fine-grained intervention can make LLM generation more accurate and stable without retraining or modifying the base model.

**Strengths:**

- The paper introduces a simple yet effective framework for real-time self-correction that can be applied to any language model without retraining or architectural changes.

- The method provides a clear mechanism for detecting and mitigating local generation errors, showing how token-level perplexity can guide correction in an interpretable way.

- Experimental results demonstrate consistent performance gains across diverse reasoning and math benchmarks, confirming the approach's robustness and practicality.

- The framework is computationally lightweight compared to multi-pass refinement methods, since it reuses the model's own activations and verifier feedback rather than generating full alternative outputs.

- The paper contributes new insights into how continuous monitoring of model uncertainty can improve stability and factual accuracy in step-by-step generation.

**Weaknesses:**

- The paper lacks a clear theoretical justification for why token-level perplexity reliably correlates with factual or reasoning errors, relying mainly on intuition and anecdotal examples (Figure 2).

- The approach introduces additional inference cost through iterative token re-evaluation and verifier calls, but the paper does not report latency, FLOPs, or wall-clock overhead to substantiate its claims of efficiency. The only quantitative evidence concerns token usage, which measures output length rather than actual computational or time efficiency.

- The method's reliance on external verifier feedback limits reproducibility and fairness, since different verifier choices or access levels could yield inconsistent results across users and model settings.

- The verifier component is insufficiently analyzed: the paper does not show how verifier quality, domain alignment, or calibration affect correction accuracy or error propagation.

- The experiments focus narrowly on reasoning and math datasets, leaving it unclear whether the method generalizes to open-ended generation, dialogue, or long-context settings.

**Questions:**

How does Once-More decide when to trigger a correction during generation, and what threshold or criterion determines whether a high-perplexity segment actually warrants intervention?

---

> ### Author Response · Authors · 2025-11-22
> **Response to Reviewer o7eJ (Part 1)**
>
> We thank the reviewer for their constructive feedback and for recognizing the effectiveness, robustness, and lightweight nature of the Once-More framework. We appreciate the reviewer's support and acknowledgment that our method offers a clear mechanism for mitigating errors without retraining. Below, we address the specific concerns raised.
>
> ---
>
>  >  ### **1. Weakness 1: Theoretical Justification for Perplexity**
>
> We acknowledge the reviewer’s concern that the link between perplexity and error is largely empirical. While there is a lack of formal theoretical proof for this assumption, substantial recent literature supports the use of entropy-based metrics as reliable signals for correctness. For instance, Yu et al. [1] uses confidence scores of current content to estimate reasoning trace quality, and Yang et al. [2] shows that local reasoning steps are sufficient for assessing trajectories. Furthermore, Wang et al. [3] demonstrates that a subset of high-entropy local tokens disproportionately influences global reasoning quality.
>
> To empirically validate this, we analyzed the relationship between intermediate unit perplexity and final response correctness. We report the distribution of the mean top-10% unit-level perplexities for correct versus incorrect responses across LiveBench, GPQA, and AIME. We observed a clear separation in distributions: incorrect responses consistently exhibit substantially higher top-10% perplexity than correct ones. This strongly supports our assumption that local uncertainty signals (perplexity)  are indicative of eventual correctness and can reliably guide verification.
>
> ---
> **Table 1: Distribution Statistics (Mean ± Std) of Highest 10% Mean Perplexity for Correct and Incorrect Responses**
>
> | Benchmark | Correct Responses  | Incorrect Responses  |
> |:----------|:------------------------------:|:--------------------------------:|
> | AIME      | 2.65 ± 0.41                    | 3.16 ± 0.28                      |
> | LiveBench | 2.71 ± 0.53                    | 3.37 ± 0.26                      |
> | GPQA      | 2.52 ± 0.45                    | 3.11 ± 0.32                      |
> ---
>
> [1] Yu et al., “DeepThink with Confidence: Filtering and Evaluating Intermediate Reasoning with Uncertainty Signals,” arXiv:2508.15260, 2025.
>
> [2] Yang et al., “Less is More: Improving LLM Reasoning with Minimal Test-Time Intervention,” arXiv:2510.13940, 2025.
>
> [3] Wang et al., “Beyond the 80/20 Rule: High-Entropy Minority Tokens Drive Effective Reinforcement Learning for LLM Reasoning,” in Advances in Neural Information Processing Systems (NeurIPS) 2025.
>
> > ### **2. Weakness 2: Latency and Wall-Clock Time.**
>
> We agree that token usage alone does not capture the full computational cost. To address this, we measured wall-clock time on an H100 server across our benchmarks. The wall-clock reported in the following table is derived from Qwen3 14B, with the following parameters: α (suppression strength) = 1, K (top-k perplexity) = 1, σ (diffusion bandwidth) =1.
>
> ---
> **Table 2: Average Wall-Clock Time to Generate One Answer by Task and Method on Qwen3-14B Model (seconds)**
>
> | Task | Raw | SelfRefine | CRITIC | Ours |
> |:-----|:---:|:----------:|:------:|:----:|
> | AIME24 | 37.94 | 128.42 | 136.92 | 124.37 |
> | AIME25 | 49.45 | 132.43 | 144.27 | 138.65 |
> | LiveBench (Reasoning) | 35.67 | 43.07 | 46.33 | 44.91 |
> | GPQA | 10.81 | 15.61 | 17.40 | 12.21 |
>
> ---
>
> Our results indicate that Once-More has a comparable runtime to Self-Refine on complex tasks (like AIME) where corrections are frequent. However, on easier tasks (like GPQA), Once-More is significantly faster because the continuous monitoring allows the model to proceed without intervention when perplexity is low, avoiding the mandatory re-generation overhead of post-hoc methods. We will include these latency comparisons in the final paper.

---

> ### Author Response · Authors · 2025-11-22
> **Response to Reviewer o7eJ (Part 2)**
>
> >  ### **3. Weakness 3 and 4: Verifier Analysis**
>
> We agree that verifier quality is a critical variable. To address reproducibility, our standard implementation uses the Producer model itself as the verifier, ensuring a self-contained and reproducible system. To address the concern regarding verifier quality, we conducted an ablation study using asymmetric Producer/verifier pairs (Qwen 4B/8B/14B). The following tables show the framework accuracy under these asymmetric settings.
>
> ---
> **Table 3: Framework Accuracy Under Producer: Qwen-14b**
>
> | Task | Qwen-14b (Verifier) | Qwen-8b (Verifier) | Qwen-4b (Verifier) | Raw model |
> |:-----|:-------------------:|:------------------:|:------------------:|:---------:|
> | AIME24 | 36.6 | 36.6 | 30.0 | 26.6 |
> | LiveBench | 52.5 | 49.5 | 46.5 | 44.0 |
> | GPQA | 55.6 | 51.5 | 50.0 | 48.0 |
>
> ---
> ---
>
> **Table 4: Framework Accuracy Under Producer: Qwen-4b**
>
> | Task | Qwen-4b (Verifier) | Qwen-8b (Verifier) | Qwen-14b (Verifier) | Raw model |
> |:-----|:------------------:|:------------------:|:-------------------:|:---------:|
> | AIME24 | 16.7 | 23.3 | 33.3 | 13.3 |
> | LiveBench | 33.0 | 38.5 | 41.0 | 20.3 |
> | GPQA | 47.5 | 52.5 | 54.0 | 43.9 |
> ---
>
> With a strong Producer (Qwen-14B), performance degrades only mildly with weaker verifiers (4B/8B), remaining substantially better than the raw baseline.
>
> With a weak Producer (Qwen-4B), increasing verifier strength (using 14B) yields massive improvements across all benchmarks.
>
> This confirms that while better verifiers yield better results, the framework functions effectively even when the verifier is weaker than the Producer.
>
>  >  ### **4. Weakness 5: Generalization to Open-Ended Generation**
>
> We would like to thank reviewer o7eJ for raising this thoughtful comment on the applicability to open-ended generation. While our primary focus was reasoning, we agree that testing on open-ended tasks is valuable. Actually, this idea of self-correction is first proposed for long sequence generation. We want the proposed once-more could reject the “off-track” units and maintain a complete story line. However, there are no open source “story generation” benchmarks or any public accepted tool to evaluate the generated story. We therefore pick popular benchmark in field of math solving, reasoning and QA answering to justify our framework.
>
> With this comment, we conducted an additional experiment on Story Generation using the WritingPrompts dataset. We evaluated coherence, creativity, logical consistency, and character consistency (scale 0–10) using GPT-5 as a judge. The results below represent the mean scores of stories generated from the first 100 prompts in the WritinPrompts dataset.
>
> ---
> **Table 5: Story Quality Evaluation by GPT-5 (Mean Scores from First 100 Prompts)**
>
> | Method | Coherence | Creativity | Logical Consistency | Character Consistency |
> |:-------|:---------:|:----------:|:-------------------:|:---------------------:|
> | Raw model | 8.15 | 8.05 | 7.30 | 8.00 |
> | SelfRefine | 7.60 | 8.40 | 7.30 | 7.50 |
> | Critic | 7.95 | 8.25 | 7.30 | 8.10 |
> | Ours | 8.20 | 8.65 | 7.40 | 8.10 |
>
> ---
>
> Once-More outperformed the raw model and Self-Refine, particularly in creativity and logical consistency. This demonstrates that the framework generalizes effectively to creative writing by maintaining narrative coherence.

---

> ### Author Response · Authors · 2025-11-22
> **Response to Reviewer o7eJ (Part 3)**
>
> > ### **5. Question 1: Perplexity Trigger**
>
> > ### **5.1 Trigger Mechanism**
>
> The decision to pause generation and inspect a unit is determined by the Unit-Level Perplexity ($PPL_{unit}$). We employ a calibration step where we compute the perplexities of answer units from a small random sample of the task data (sample sizes were 7 for AIME 24/25 and 50 for LiveBench and GPQA-Diamond). We define a hyper parameter η as the target verification rate. The triggering threshold is set to the top η-th percentile of these calibrated perplexity scores. During inference, any unit with $PPL_{unit}$ exceeding this threshold triggers the verification process. In our experiment, we set hyper parameter threshold η=25%
>
> > ### **5.2 Determining Intervention**
>
> It is important to clarify that exceeding the perplexity threshold only triggers the inspection, it does not force a correction. The determination of whether intervention is actually needed is the job of the Verifier. If the Verifier rejects the triggered unit, the framework initiates the logit-redistribution and regeneration process.
>
> > ### **5.3 Threshold Selection Analysis**
>
> To select the optimal criterion η, we analyzed the trade-off between computational overhead and error detection capability. Assuming the Verifier as an oracle on the GPQA benchmark (Qwen-14B), we measured the False Positive Rate (units triggered by high perplexity but found correct by the verifier) and False Negative Rate (erroneous units that failed to trigger the threshold).
>
> ---
> **Table 6: False Positive and False Negative Rates at Different Perplexity Thresholds (GPQA, Qwen-14B)**
>
> | Metric | η = 12.5 | η = 25 | η = 50 |
> |:-------|:--------:|:------:|:------:|
> | False positive rate (%) | 39.6 | 45.8 | 51.9 |
> | False negative rate (%) | 80.1 | 48.6 | 32.4 |
> ---
>
> The results demonstrate a clear trade-off. A strict threshold (η = 12.5%) yields the lowest false positive rate. This means most triggered units are indeed problematic. However, it suffers from a high false negative rate (80.1%) and fails to detect a significant number of errors. Conversely, a looser threshold (η = 50%) results in a higher false positive rate but achieves the lowest false negative rate. Although the latter requires more computational overhead, it is significantly more capable of identifying problematic answer units.
>
> We take one step further to analysis the relationship between perplexity threshold η and hyper parameter K (top-K for perplexity estimation). Since the parameter K (top-K for perplexity estimation) directly defines unit perplexity and consequently impacts the threshold ratio η, we conducted a joint ablation study of these two parameters. This study was performed using the Qwen-14B model on AIME24 and GPQA-Diamond. All other parameters were fixed as follows: α (suppression strength) = 1, and σ (diffusion bandwidth) =1, τ (distance decay factor) = 1. The following tables contain accuracy and total wall-clock time to finish the entire benchmark.
>
> ---
> **Table 7: AIME24 Results - Accuracy and Wall-Clock Time**
>
> |     | η = 12.5% |          | η = 25% |          | η = 50% |          |
> |:---:|:---------:|:--------:|:-------:|:--------:|:-------:|:--------:|
> |     | Acc.      | Time (s) | Acc.    | Time (s) | Acc.    | Time (s) |
> | K=1 | 36.6      | 1693     | 36.6    | 3579     | 36.6    | 6448     |
> | K=5 | 36.6      | 2448     | 33.3    | 4133     | 36.6    | 6103     |
> | K=10| 33.3      | 2397     | 33.3    | 3142     | 36.6    | 5954     |
> | K=15| 30.0      | 2929     | 33.3    | 2851     | 33.3    | 7088     |
>
> ---
>
> ---
> **Table 8: GPQA Results - Accuracy and Wall-Clock Time**
>
> |     | η = 12.5% |          | η = 25% |          | η = 50% |          |
> |:---:|:---------:|:--------:|:-------:|:--------:|:-------:|:--------:|
> |     | Acc.      | Time (s) | Acc.    | Time (s) | Acc.    | Time (s) |
> | K=1 | 51.0      | 1987     | 54.5    | 3640     | 61.6    | 4953     |
> | K=5 | 47.5      | 2448     | 55.5    | 3848     | 55.0    | 4424     |
> | K=10| 49.0      | 2397     | 49.5    | 2098     | 53.5    | 4244     |
> | K=15| 48.4      | 2473     | 49.5    | 2486     | 55.0    | 3412     |
>
> ---
>
> The result further verified our found trade-off or false positive/negative rate. For small K (1 or 5), changing η primarily impacts runtime rather than accuracy. A looser threshold (η = 50%) increases verification calls and computation time, while a stricter threshold (η = 12.5%) improves efficiency but reduce performance. In scenarios where computational resources are unconstrained, one can safely loosen the threshold by selecting a larger η value to maximize model performance.
>
>
> > ### **6. End of response**
>
> We sincerely thank the reviewer for the thoughtful feedback that has strengthened our work. We will update the manuscript shortly to incorporate these new ablation studies and analyses. We appreciate your time and effort in reviewing our comprehensive response.

---

### Official Review · Reviewer_Gzea · 2025-11-01

**Soundness:** 2
**Presentation:** 3
**Contribution:** 3
**Rating:** 4
**Confidence:** 3

**Summary:**

This work proposes a model-agnostic framework for continuous self-correction for LLMs at inference time. Unlike contemporary post-hoc iterative refinement methods that wait for complete outputs, authors propose to intervene at a more granular level of a "unit" (sentences, paragraphs or code blocks) using two key mechanisms: (1) perplexity monitoring to detect potential errors (2) fusing logit redistribution with verifier feedback (LLM-based or symbolic) to guide regeneration away from failed attempts.

The work motivates the necessity to intervene early to prevent the propagation of errors in early steps to the final outputs. The continuous intervention approach is conceptually interesting, and to the best of my knowledge, the combination of perplexity signals with external feedback mechanisms is novel. The proposed method achieves strong empirical results on mathematical reasoning (AIME), the reasoning subset of LiveBench, and GPQA compared to Self-Refine and Critic.

However, the work suffers from several practical limitations, including complex hyperparameter setup, several missing details, and evaluation being limited to reasoning tasks.

**Strengths:**

- Novel Combination of Signals: The proposed method uniquely integrated unit-level perplexity with external verifier feedback for continuous guidance, correcting the wrong steps in the early stages.
- Model-Agnostic and Training Free: The framework shows strong performance despite being an inference-time technique. The setup is model agnostic and thus can be broadly applied.
- Token Efficiency Gains: The proposed method shows gains in generated token efficiency while achieving better performance.
- Scaling Behaviour: The Authors show that the framework benefits more from larger models, suggesting it effectively leverages richer model representations.

**Weaknesses:**

- The authors provide no ablation for the selection of the "unit". Unit granularity plays a pivotal role in the proposed method; however, it is not clear how the performance changes with the changing definition of the unit.
- The core assumption of the paper is that each unit can be verified without looking at the future unit. A discussion on this is warranted.
- The method introduces numerous hyperparameters (K for top-K, $\alpha$ for suppression strength, $\beta$ for redistribution sharpness, $\sigma$ for diffusion bandwidth, $\tau$ and $\gamma$ for distance decay, and $c$ for truncation radius) with minimal justification. Section 4.1 only provides the value for K, $\alpha$, $\beta$ without sensitivity analysis. The authors provide no ablation experiments to study the effects of various hyperparameters or how they were selected.
- Perplexity threshold is calibrated using empirical quantiles on a held-out set of prompts. However, no further details are provided, such as how many prompts were used for calibration or the value of verification rate $\eta$. Similar to other hyperparameters, no sensitivity analysis is done for this value.
- Authors provide a computational cost analysis in the form of generation token efficiency. However, they do not provide information on wall-clock time. Depending on the selection of units, the wall clock time may exceed that of contemporary methods. I believe an analysis of the number of verifier calls, wall-clock time will make the claims stronger.
- It is not clear if the proposed method works for an open-ended generation task. Authors limit their experiments to reasoning tasks where units can be sufficiently defined. However, for open-ended tasks such as story generation, the definition of a unit and the core assumption (each unit can be verified without access to future steps) may break down.

**Questions:**

- Verifier Quality Analysis: What happens when a stronger verifier is used in an asymmetric setting? I understand this method proposes to improve the self-correction capabilities; however, seeing that the method generalizes well with scale, it may be worth showing sample results in an asymmetric setting with a stronger verifier.
- Hyperparameter Sensitivity: Please provide sensitivity analysis for all hyperparameters. Are the results robust to these choices, or does performance degrade significantly with the choice of other hyperparameter values? At the very least, please provide values for all hyperparameters that were used for experiments.
- Calibration Data Requirements: How many held-out examples are needed for reliable $P_{th}$ calibration? How does performance degrade if calibration and test distributions differ?
- Computational Overhead: What is the average wall-clock time increase compared to raw generation, and chosen baseline method?
- Perplexity-Error Correlation: What is the actual correlation between $PPL_{unit} > P_{th}$  and the presence of errors in your datasets? What are the false positive/negative rates? How does wrong perplexity calibration affect the downstream performance?
- Verifier Disagreement: How is verifier disagreement handled in the case of multiple verifiers? Does the framework support weighted voting or consensus mechanisms?
- Complexity of Logit redistribution: Have the authors tried simpler logit penalties/redistribution strategies? What is the comparison with a simple token penalty based on perplexity? Are all the components, such as distance decay, diffusion to prevent overlocalization, important? An ablation of these components would be appreciated.

---

> ### Author Response · Authors · 2025-11-22
> **Response to Reviewer Gzea (Part 1)**
>
> We sincerely thank the reviewer for the time and thoughtful effort dedicated to evaluating our work. We appreciate the recognition of our methodological novelty. In response to the reviewer's insightful concerns, we have conducted additional ablation studies and expanded analyses to directly address each weakness and question raised.
>
> ---
>
>  >  ### **1. Weakness 1: Unit Granularity Ablation**
>
> We thank the reviewer for this insightful comment. We agree that an ablation study on unit length is necessary to understand the stability of the Once-More framework. We conducted experiments on the Qwen3-14B model across three benchmarks: AIME24, LiveBench, and GPQA-Diamond. Other parameters were set to default: K (top-k perplexity) = 1, α (suppression strength) = 1, σ (diffusion bandwidth) = 1 and τ (distance decay factor) = 1.
>
> ---
> **Table 1: Small Unit Length Ablation (Mean Accuracy ± Std Dev over 3 runs)**
>
> | Benchmark | 1 sentence | 2 sentences | 4 sentences |
> |:----------|:----------:|:-----------:|:-----------:|
> | AIME24 | 36.6 ± 3.3 | 34.3 ± 1.9 | 35.5 ± 1.9 |
> | Livebench | 52.3 ± 1.2 | 52.1 ± 1.0 | 52.3 ± 1.2 |
> | GPQA-D | 55.6 ± 1.8 | 55.5 ± 1.3 | 56.1 ± 1.5 |
>
> ---
>
> We further conducted an ablation study on larger unit lengths:
>
> ---
> **Table 2: Large Unit Length Ablation (Mean Accuracy ± Std Dev over 3 runs)**
>
> | Benchmark | 32 sentences | 64 sentences | 128 sentences |
> |:----------|:------------:|:------------:|:-------------:|
> | AIME24 | 30.0 ± 2.7 | 26.7 ± 2.7 | 26.7 ± 2.7 |
> | Livebench | 49.8 ± 1.7 | 50.3 ± 1.3 | 45.3 ± 0.6 |
> | GPQA-D | 49.7 ± 1.0 | 49.3 ± 0.3 | 50.0 ± 0.7 |
>
> ---
>
> The results indicate no significant performance difference for small unit lengths (<4 sentences). This demonstrates that the framework is robust under fine-grained segmentation. However, when the unit length increases significantly (>32 sentences), performance drops across all three benchmarks, and converge to the performance of the raw model (without Once-More) at a unit length of 128.
>
> > ### **2. Weakness 2: Verification Assumption**
>
> We acknowledge the reviewer’s concern. Verifying a unit without explicit conditioning on future tokens is indeed an assumption. However, recent literature suggests that evaluating partial content via entropy-related metrics provides a reliable signal for response quality. For instance, Yu et al. [1] uses confidence scores of current content to estimate reasoning trace quality, and Yang et al. [2] shows that local reasoning steps are sufficient for assessing trajectories. Furthermore, Wang et al. [3] demonstrates that a subset of high-entropy local tokens disproportionately influences global reasoning quality. To empirically validate this, we analyzed the relationship between intermediate unit perplexity and final response correctness. We also report the distribution of the mean top-10% unit-level perplexities for correct versus incorrect responses across LiveBench, GPQA, and AIME. We observed a clear separation in distributions: incorrect responses consistently exhibit substantially higher top-10% perplexity than correct ones. This strongly supports our assumption that local uncertainty signals (perplexity)  are indicative of eventual correctness and can reliably guide verification.
>
> ---
> **Table 3: Distribution Statistics (Mean ± Std) of Highest 10% Mean Perplexity for Correct and Incorrect Responses**
>
> | Benchmark | Correct Responses  | Incorrect Responses  |
> |:----------|:------------------------------:|:--------------------------------:|
> | AIME      | 2.65 ± 0.41                    | 3.16 ± 0.28                      |
> | LiveBench | 2.71 ± 0.53                    | 3.37 ± 0.26                      |
> | GPQA      | 2.52 ± 0.45                    | 3.11 ± 0.32                      |
> ---
>
> [1] Yu et al., “DeepThink with Confidence: Filtering and Evaluating Intermediate Reasoning with Uncertainty Signals,” arXiv:2508.15260, 2025.
>
> [2] Yang et al., “Less is More: Improving LLM Reasoning with Minimal Test-Time Intervention,” arXiv:2510.13940, 2025.
>
> [3] Wang et al., “Beyond the 80/20 Rule: High-Entropy Minority Tokens Drive Effective Reinforcement Learning for LLM Reasoning,” in Advances in Neural Information Processing Systems (NeurIPS) 2025.

---

> ### Author Response · Authors · 2025-11-22
> **Response to Reviewer Gzea (Part 2)**
>
> > ### **3. Weakness 3: Hyperparameters**
>
> We thank the reviewer for this suggestion. In practice, the redistribution sharpness β and distance decay norm γ have negligible impact on the model’s performance. These parameters were included primarily to present a more formal and general formulation of the redistribution process. The parameter β works as temperature controls for alternative generation style and γ just controls the first norm distance or second norm distance used in the distance decay function, they have minimal effect on the accuracy or the behaviour of our framework.
>
> To avoid unnecessary complexity, we will update the manuscript to fix them to default values. The only hyperparameters that meaningfully affect performance are K (top-K for perplexity estimation), α (suppression strength), σ (diffusion bandwidth) and τ (distance decay factor).
>
> > ### **3.1 Suppression Strength (α)**
>
> The ablation study on α (suppression strength) is conducted by Qwen-14B model on AIME24, Livebench and GPQA-Diamond. Due to the time constraint, the ablation study only shows the performance of the model in one run. All other parameters are chosen as: K (top-k perplexity) = 1, σ (diffusion bandwidth) =1 and τ (distance decay factor) =1
>
> ---
> **Table 4: Ablation on Suppression Strength (α). Accuracy from single run**
>
> | Benchmark |  α = 0.1  |  α = 0.5  |  α = 1  | α = 1.5 | α = 2 |
> |:----------|:-------:|:-------:|:-----:|:-------:|:-----:|
> | AIME24 | 30.0 | 33.3 | 36.6 | 33.3 | 36.6 |
> | Livebench | 50.5 | 51.5 | 52.0 | 52.5 | 51.0 |
> | GPQA-D | 49.5 | 52.5 | 54.5 | 54.5 | 54.0 |
> ---
>
> Performance is stable for α > 1. However, a small strength (α = 0.1) dilutes the suppression effect, significantly reducing performance.
>
> > ### **3.2 Diffusion Bandwidth (σ)**
>
> The ablation study on σ (diffusion bandwidth) is conducted by Qwen-14B model on AIME24, Livebench and GPQA-Diamond. Due to the time constraint, the ablation study only shows the performance of the model in one run. All other parameters are chosen as: K (top-k perplexity) = 1, α (suppression strength) = 1, and τ (distance decay factor) =1.
>
> ---
> **Table 5: Ablation on Diffusion Bandwidth (σ). Accuracy from single run**
>
> | Benchmark | σ = 0.1 | σ = 1 | σ = 5 | σ = 10 |
> |:----------|:-------:|:-----:|:-----:|:------:|
> | AIME24 | 33.3 | 36.3 | 26.6 | 30.0 |
> | Livebench | 52.5 | 55.0 | 48.5 | 47.5 |
> | GPQA-D | 50.5 | 52.5 | 49.0 | 49.5 |
> ---
>
> The results show that a large σ(>=5) spreads the suppression too broadly. It dilutes the suppression effect on the target token and behaves similarly to using a very small suppression strength (α). Conversely, a very small σ confines suppression only to the current token, making the adjustment too localized and also degrading accuracy.
>
> > ### **3.3 Decay Factor (τ)**
>
> The ablation study on τ (distance decay factor) is conducted by Qwen-14B model on AIME24, Livebench and GPQA-Diamond. The granularity of τ is set to 0.01 (extreme small τ representing decay to 0, suppression only works on target token in this specific position), 0.5, 1, 1.5 and 100 (extreme large τ representing no decay, the target token will be equally suppressed regardless its position). All other parameters are chosen as: α (suppression strength) = 1, K (top-k perplexity) = 1, and σ (diffusion bandwidth) =1.
>
> ---
> **Table 6: Ablation on Distance Decay Factor (τ). Accuracy from single run**
>
> | Benchmark | τ = 0.01 | τ = 0.5 | τ = 1 | τ = 1.5 | τ = 100 |
> |:----------|:--------:|:-------:|:-----:|:-------:|:-------:|
> | AIME24 | 23.3 | 36.6 | 36.6 | 36.6 | 30.3 |
> | Livebench | 43.5 | 52.5 | 52.5 | 52.0 | 50.5 |
> | GPQA-D | 45.5 | 54.5 | 55.0 | 55.0 | 52.5 |
> ---
>
> The ablation results show a very interesting phenomenon. Without distance decay (an extremely small τ = 0.01), model performance decreases significantly, becoming even worse than the raw model baseline. This is because, in this setting, the OnceMore framework only suppresses the target token if it is in the exact same position as in the previously rejected sequence. The model sometimes finds a trick to regenerate. Assuming this is the previously rejected unit:
> ```
> This sequence contains error and therefore got rejected
> ```
> It first generates a few other words and then repeats exactly the same sequence as the previously rejected one, just like:
> ```
> Got it! This sequence contains error and therefore got rejected
> ```
> Since the position is mismatched due to these beginning words, the later regenerated sequence is not suppressed at all. However, this is subsequently rejected by the reviewer, causing the model to get stuck in a loop until the maximum token limit is reached.
>
> The ablation for K (top-K for perplexity estimation) is shown in the next section together with perplexity threshold η to address another concern.

---

> ### Author Response · Authors · 2025-11-22
> **Response to Reviewer Gzea (Part 3)**
>
> > ### **4. Weakness 4: Perplexity threshold**
>
> Thank you for raising this detailed and important comment regarding our perplexity-threshold calibration. The perplexity threshold was set to the top 25th percentile (η = 25%) of answer units from a random sample of questions (sample sizes were 7 for AIME 24/25, and 50 for LiveBench and GPQA-Diamond). Since the parameter K (top-K for perplexity estimation) directly defines unit perplexity and consequently impacts the threshold ratio η, we conducted a joint ablation study of these two parameters. This study was performed using the Qwen-14B model on AIME24 and GPQA-Diamond. All other parameters were fixed as follows: α (suppression strength) = 1, and σ (diffusion bandwidth) =1, τ (distance decay factor) = 1. The following tables contain accuracy and total wall-clock time to finish the entire benchmark.
>
> ---
> **Table 7: AIME24 Results - Accuracy and Wall-Clock Time**
>
> |     | η = 12.5% |          | η = 25% |          | η = 50% |          |
> |:---:|:---------:|:--------:|:-------:|:--------:|:-------:|:--------:|
> |     | Acc.      | Time (s) | Acc.    | Time (s) | Acc.    | Time (s) |
> | K=1 | 36.6      | 1693     | 36.6    | 3579     | 36.6    | 6448     |
> | K=5 | 36.6      | 2448     | 33.3    | 4133     | 36.6    | 6103     |
> | K=10| 33.3      | 2397     | 33.3    | 3142     | 36.6    | 5954     |
> | K=15| 30.0      | 2929     | 33.3    | 2851     | 33.3    | 7088     |
>
> ---
>
> For small K (1 or 5), changing η primarily impacts runtime rather than accuracy. A looser threshold (η = 50%) increases verification calls and computation time, while a stricter threshold (η = 12.5%) improves efficiency. Large K values generally degrade accuracy. We observe similar trends on GPQA.
>
> ---
> **Table 8: GPQA Results - Accuracy and Wall-Clock Time**
>
> |     | η = 12.5% |          | η = 25% |          | η = 50% |          |
> |:---:|:---------:|:--------:|:-------:|:--------:|:-------:|:--------:|
> |     | Acc.      | Time (s) | Acc.    | Time (s) | Acc.    | Time (s) |
> | K=1 | 51.0      | 1987     | 54.5    | 3640     | 61.6    | 4953     |
> | K=5 | 47.5      | 2448     | 55.5    | 3848     | 55.0    | 4424     |
> | K=10| 49.0      | 2397     | 49.5    | 2098     | 53.5    | 4244     |
> | K=15| 48.4      | 2473     | 49.5    | 2486     | 55.0    | 3412     |
>
> ---
>
> > ### **5. Weakness 5:  Computational cost analysis**
>
> Thank you for highlighting the need for a more complete computational cost analysis; we agree that reporting wall-clock time is crucial for a fair comparison. The wall-clock reported in the following table is derived from Qwen3 14B, with the following parameters: α (suppression strength) = 1, K (top-k perplexity) = 1, σ (diffusion bandwidth) =1. The experiments are conducted on an H100 server
>
> ---
> **Table 9: Average Wall-Clock Time to Generate One Answer by Task and Method on Qwen3-14B Model (seconds)**
>
> | Task | Raw | SelfRefine | CRITIC | Ours |
> |:-----|:---:|:----------:|:------:|:----:|
> | AIME24 | 37.94 | 128.42 | 136.92 | 124.37 |
> | AIME25 | 49.45 | 132.43 | 144.27 | 138.65 |
> | LiveBench (Reasoning) | 35.67 | 43.07 | 46.33 | 44.91 |
> | GPQA | 10.81 | 15.61 | 17.40 | 12.21 |
>
> ---
> Experiments result shows that Once-More has a comparable runtime to Self-Refine  on AIME tasks while requiring significantly less time on easier tasks like GPQA, as fewer corrections are triggered.
>
>
> > ### **6. Weakness 6: Open-ended generation task**
>
> Thank you for raising this thoughtful comment on the applicability to open-ended generation; we share the reviewer’s intuition that extending our framework beyond structured reasoning tasks. This idea of self-correction is first proposed for long sequence generation. We want the proposed once-more could reject the “off-track” units and maintain a complete story line. However, there are no open source “story generation” benchmarks or any public accepted tool to evaluate the generated story. We therefore pick popular benchmark in field of math solving, reasoning and QA answering to justify our framework.
>
> We conducted additional experiments on story generation using prompts from the WritingPrompts dataset. We employed the Qwen-14B model with the default hyper parameters set to 1. Story quality was evaluated by GPT-5 on a scale of 0–10 across four dimensions: coherence, creativity, logical consistency, and character consistency. The results below represent the mean scores of stories generated from the first 100 prompts in the dataset.
>
> ---
> **Table 10: Story Quality Evaluation by GPT-5 (Mean Scores from First 100 Prompts)**
>
> | Method | Coherence | Creativity | Logical Consistency | Character Consistency |
> |:-------|:---------:|:----------:|:-------------------:|:---------------------:|
> | Raw model | 8.15 | 8.05 | 7.30 | 8.00 |
> | SelfRefine | 7.60 | 8.40 | 7.30 | 7.50 |
> | Critic | 7.95 | 8.25 | 7.30 | 8.10 |
> | Ours | 8.20 | 8.65 | 7.40 | 8.10 |
>
> ---
> Our method achieves highest score in all four criteria compared with baseline methods.

---

> ### Author Response · Authors · 2025-11-22
> **Response to Reviewer Gzea (Part 4/5)**
>
> > ### **7. Question 1: Asymmetric settings of verifier**
>
> Thank you for the suggestion. We agree that asymmetric settings are important, and we have already conducted experiments using both strong-producer / weak-verifier and weak-producer / strong-verifier configurations. The following tables show the framework accuracy under these asymmetric settings with default hyper parameters set to 1.
>
> ---
> **Table 11: Framework Accuracy Under Producer: Qwen-14b**
>
> | Task | Qwen-14b (Verifier) | Qwen-8b (Verifier) | Qwen-4b (Verifier) | Raw model |
> |:-----|:-------------------:|:------------------:|:------------------:|:---------:|
> | AIME24 | 36.6 | 36.6 | 30.0 | 26.6 |
> | LiveBench | 52.5 | 49.5 | 46.5 | 44.0 |
> | GPQA | 55.6 | 51.5 | 50.0 | 48.0 |
>
> ---
> ---
>
> **Table 12: Framework Accuracy Under Producer: Qwen-4b**
>
> | Task | Qwen-4b (Verifier) | Qwen-8b (Verifier) | Qwen-14b (Verifier) | Raw model |
> |:-----|:------------------:|:------------------:|:-------------------:|:---------:|
> | AIME24 | 16.7 | 23.3 | 33.3 | 13.3 |
> | LiveBench | 33.0 | 38.5 | 41.0 | 20.3 |
> | GPQA | 47.5 | 52.5 | 54.0 | 43.9 |
> ---
>
> With a strong Producer (Qwen-14B), performance degrades only mildly with weaker verifiers (4B/8B), remaining substantially better than the raw baseline.
>
> With a weak Producer (Qwen-4B), increasing verifier strength (using 14B) yields massive improvements across all benchmarks
>
>
> > ### **8. Question 2: Hyperparameters sensitivity**
>
> Thank you for emphasizing the importance of hyperparameter sensitivity; we agree that understanding the robustness of our method to these choices is essential. These are the hyper parameter values we used for our experiments: α (suppression strength) = 1, K (top-k perplexity) = 1, η (perplexity threshold) = 25%, σ (diffusion bandwidth) =1 and unit length = 1 and τ (distance decay factor) = 1. The ablation study of these hyper parameters are already detailed in previous sections 3 and 4.
>
> > ### **9. Question 3: Perplexity Threshold**
>
> Thank you for pointing out the need to clarify our calibration procedure. The perplexity threshold was set to the top 25th percentile (η = 25%) of answer units from randomly sampled questions (sample sizes were 7 for AIME 24/25 and 50 for LiveBench and GPQA-Diamond). According to the ablation study in Section 4, a stricter threshold (lower η) slightly degrades model performance, as it causes the verifier to overlook some problematic answer units (false negatives). Conversely, a looser threshold (higher η) does not negatively impact performance accuracy but does increase computational time. Therefore, in scenarios where computational resources are unconstrained, one can safely loosen the threshold by selecting a larger η value to maximize model performance.
>
> > ### **10. Question 4: Computational Overhead**
>
> Thank you for highlighting the need for a more complete computational cost analysis. Please check the wall-clock time comparison on previous section 5.
>
>
> > ### **11. Question 5: Perplexity-Error Correlation**
>
> Thank you for your insightful question, we actually like this question very much. We acknowledge that we cannot guarantee a perfect correlation between Perplexity > Threshold and the presence of an error. Therefore, we propose using perplexity solely as a trigger for the verifier to inspect specific answer units. As discussed in Section 2, our core assumption is simply that high perplexity in answer units correlates with the final correctness of the answer. To quantify this, we treat our Verifier model as an oracle that can accurately detect problematic units. We define the False Positive Rate (FPR) as cases where perplexity > η (triggering the verifier), but the verifier finds no error. Conversely, the False Negative Rate (FNR) refers to problematic units (judged by the verifier) that failed to trigger the perplexity threshold η.We conducted an experiment on GPQA using the Qwen-14B model to estimate these rates across different thresholds (η):
>
> ---
> **Table 14: False Positive and False Negative Rates at Different Perplexity Thresholds (GPQA, Qwen-14B)**
>
> | Metric | η = 12.5 | η = 25 | η = 50 |
> |:-------|:--------:|:------:|:------:|
> | False positive rate (%) | 39.6 | 45.8 | 51.9 |
> | False negative rate (%) | 80.1 | 48.6 | 32.4 |
> ---
>
> The results demonstrate a clear trade-off. A strict threshold (η = 12.5%) yields the lowest false positive rate, meaning most triggered units are indeed problematic; however, it suffers from a high false negative rate (80.1%), failing to detect a significant number of errors. Conversely, a looser threshold (η = 50%) results in a higher false positive rate but achieves the lowest false negative rate. Although the latter requires more computational overhead, it is significantly more capable of identifying problematic answer units.

---

> ### Author Response · Authors · 2025-11-22
> **Response to Reviewer Gzea (Part 5/5)**
>
> > ### **12. Question 6: Verifier Disagreement**
>
> Thank you for raising this insightful question about handling disagreement among multiple verifiers. Our proposal of multi-verfiers is to let each verifier focus on different aspects to check answer units. An unit is regenerated if any verifier rejects it, and the feedback from all verifiers will be taken. With our current choice of using the Producer itself as the verifier, we found that applying this multi-verifier method did not provide significant improvement compared to using a single verifier that accounts for all aspects. However, it did impose a huge burden on computational resources. We will mention this multi-verifier concept in our revised manuscript to suggest potential directions for other verifier schemes.
>
>
> > ### **13. Question 7: Complexity of Logit redistribution**
>
> We would like to thank the reviewer to propose this critical concern. The simplest redistribution strategy is to suppress token only by location without distance decay and token penalty diffusion. By setting τ (distance decay factor) = 0.01, σ (diffusion bandwidth) =0.1, we can simulate this simple redistribution strategy. From our ablation study on section 3.2 and 3.3, we can observe that this strategy will significant reduce the performance of OnceMore model. The Producer will find a trick way to regenerate: just simply add a few new tokens in the beginning index to mismatch the location of each token and regenereate the same sequence as previously rejected one. That is the reason why we introduce distance decay and diffusion to suppress the token with high perplexity
>
>
> > ### **Ending of this response**
>
>
> We sincerely thank Reviewer Gzea for the valuable and insightful comments that have significantly strengthened our work. We appreciate the time and effort dedicated to reviewing this comprehensive response. We will revise our manuscript to incorporate these experimental results and expand the appendix accordingly.

---

### Author Response · Authors · 2025-11-28
**Comment to All Reviewers**

We sincerely thank all reviewers for their thorough and constructive feedback. Your insightful comments have significantly strengthened the quality and clarity of our work.

We have now uploaded a revised manuscript that incorporates all the feedback received during the review process. In addition to addressing specific technical concerns, we have conducted a further round of thorough proofreading to correct grammatical errors and polish the writing throughout the paper.

Key updates to the manuscript include:

* **Statistical Rigour (Standard Deviations):** To address concerns regarding the variability of stochastic methods, we have updated our results to report the mean accuracy $\pm$ standard deviation over three independent runs. This is now reflected in the main results (**Table 1**) and the new unit length ablation study (**Table 4**).
* **Perplexity Distribution Analysis (Figure 3):** We added a new figure visualizing the distributions of mean top-10% unit-level perplexity for correct versus incorrect responses. This empirically validates our assumption that local uncertainty signals reliably correlate with generation errors.
* **Expanded Ablation Studies (Table 4 & Appendix A.5):** We have added comprehensive analyses of key hyperparameters. This includes a new **Table 4** reporting the unit length ablation study, as well as detailed ablations in **Appendix A.5** for suppression strength ($\alpha$), distance decay ($\tau$), diffusion bandwidth ($\sigma$), and the perplexity threshold ($\eta$).
* **Verifier Analysis (Table 5):** We included results from asymmetric generator/verifier experiments (e.g., strong producer with weak verifier) to demonstrate the framework's robustness across different model capabilities.
* **Efficiency Metrics (Figure 4):** We expanded our efficiency analysis to include **wall-clock time** comparisons alongside token usage, providing a more complete picture of the computational overhead.
* **Clarifications & Corrections:** We have refined the methodology section for clarity and expanded the **Conclusion** (Section 5) to explicitly address limitations regarding error tracing, false negatives, and verifier dependency.

We invite you to review these updates along with our detailed responses to your specific comments. We remain available to address any further questions or concerns you may have.

Sincerely,

The Authors

---

### Author Response · Authors · 2025-12-02
**Summary of Reviewer Consensus and Rebuttal Updates**

Dear Area Chair,

In light of the recent announcement by the conference organizer, we provide this summary to assist in your assessment. We wish to summarize the consensus regarding the paper's strengths and the extensive new comparative data generated during the rebuttal.

## 1. Important Note: Score Discrepancy

We would like to draw your attention to the final comment from **Reviewer bA69** ("Reviewer Answer", Nov 26). The reviewer explicitly stated:

> **"I have raised my score accordingly."**

However, the system remained displaying their score as **2 (Reject)** (even before the revert). Given that this reviewer acknowledged our "extensive ablations" and "strengthened technical contribution" in that same message, we believe the numerical score update was likely a technical oversight. We respectfully ask that you evaluate the submission with the understanding that this reviewer intended to improve their score.

## 2. Consensus on Novelty, Timeliness, and Practicality

We are encouraged that **all four reviewers** independently confirmed the novelty, timeliness, and practical value of the proposed *Once-More* framework in their initial reviews:

* **Reviewer Gzea:** Highlighted the work as **"conceptually interesting"** and praised the **"novel combination"** of perplexity signals with external feedback.
>**"The continuous intervention approach is conceptually interesting, and to the best of my knowledge, the combination of perplexity signals with external feedback mechanisms is novel."**
* **Reviewer o7eJ:** Described the framework as **"simple yet effective"**, noting it provides **"consistent performance gains"** and is **"computationally lightweight"**.
> **"The paper contributes new insights into how continuous monitoring of model uncertainty can improve stability and factual accuracy in step-by-step generation."**
* **Reviewer 5kHm:** Expressed that the idea is **"timely and practically relevant"** for improving LLM reliability and recognized the approach as **"sensible and practical"**.
>**"Results show consistent improvements and a token-efficiency advantage over some baselines. The idea is timely and practically relevant for improving reliability of LLM generation."**
* **Reviewer bA69:** Acknowledged the **"novel algorithm"** and stated the general framework is a **"great contribution"** with **"observable gains"**.
>**"the general framework is a great contribution, especially when driven by the uncertainty estimator, which is a powerful tool for analyzing correctness and is believed to be a bottleneck in self-correcting papers"**

## 3. Addressing Concerns via Extensive Comparisons & Ablation Studies

The primary concerns raised during the review process focused on the need for more extensive comparative data and sensitivity analyses. We have thoroughly addressed these by generating significant new empirical evidence:

* **Comprehensive Ablation Studies (Addressing Gzea, 5kHm, bA69):** To prove the robustness of our framework, we conducted exhaustive sensitivity analyses on all key hyperparameters. This includes:
    * **Component Ablations:** Suppression strength ($\alpha$), diffusion bandwidth ($\sigma$), and distance decay ($\tau$).
    * **Perplexity Sensitivity:** Joint ablation of the perplexity threshold ($\eta$) and top-$K$ estimation.
    * **Unit Granularity:** Experiments varying unit length (1 vs. 2 vs. 4 vs. 32 sentences) demonstrating framework stability under fine-grained segmentation.

* **Efficiency vs. Self-Consistency (Addressing bA69):** We compared *Once-More* against Self-Consistency ($k=10$). On GPQA, our method achieved **higher accuracy (55.6% vs 51.5%)** while using **~10x fewer tokens** and being **~5x faster**. This addresses concerns about computational overhead vs. standard sampling methods.

* **Wall-Clock Latency (Addressing Gzea, o7eJ, 5kHm):** We provided wall-clock time comparisons showing *Once-More* is comparable to or faster than baselines like Self-Refine on easier tasks, disproving concerns about latency overhead.

* **Generalization to Creative Tasks (Addressing Gzea, o7eJ):** To demonstrate the framework is model-agnostic beyond math/reasoning, we conducted a new **Story Generation** experiment (WritingPrompts). *Once-More* outperformed baselines in Creativity and Logical Consistency as evaluated by GPT-5.

* **Robustness & Asymmetric Verifiers (Addressing Gzea, 5kHm):** We added ablation studies using asymmetric producer/verifier pairs (e.g., Qwen-4B verifying Qwen-14B). The results confirm practical improvements even when using smaller, efficient verifiers.

## Conclusion

The reviewers have reached a consensus on the work's novelty and practical strengths. The major concern (needing additional comparative data) has been met with extensive new experiments demonstrating superior efficiency and robustness. We hope the Area Chair will consider these significant updates and the intended score improvement from Reviewer bA69.

Sincerely,

The Authors

---

### Meta-Review · Area_Chair_yaT3 · 2026-01-07

**Summary:**

The paper introduces Once-More, a model-agnostic framework for continuous self-correction during large language model (LLM) generation. The central idea is to monitor perplexity at the unit or token level and invoke verifier feedback to guide logit redistribution, thereby correcting errors before they propagate. This fine-grained intervention contrasts with post-hoc refinement methods and is evaluated on several reasoning and mathematical benchmarks.

**Reviewer Concerns:**

- Experimental Scope and Baselines: Several reviewers note that the evaluation is limited to reasoning and math tasks. The absence of broader domains (e.g., open-ended generation, dialogue) raises questions about generalizability. Reviewer 4 in particular criticizes the choice and quality of baselines, suggesting that stronger or more diverse comparisons are needed.
- Hyperparameter Complexity and Sensitivity: Multiple reviewers highlight the large number of hyperparameters introduced, with insufficient justification or sensitivity analysis. The lack of clarity on calibration procedures and unit segmentation further weakens reproducibility.
- Verifier Analysis: The role and quality of verifiers are underexplored. Reviewers ask how verifier choice affects performance, how disagreements are handled, and whether weaker verifiers degrade results significantly.
- Efficiency Claims: While token usage is reported, there is no analysis of wall-clock time, latency, or computational overhead. This omission undermines claims of efficiency.

**Reviewer Scores:**

Reviewer bA69 has stated to raise the score. Reviewer 5kHm may have a chance to raise the score.

---

### Decision · Program_Chairs · 2026-01-26

Accept (Poster)